# Cooperation-based concept formation in male bottlenose dolphins

Stephanie L. King [1,2✉], Richard C. Connor[3], Michael Krützen[4] & Simon J. Allen[1,2,4]

In Shark Bay, Western Australia, male bottlenose dolphins form a complex nested alliance hierarchy. At the first level, pairs or trios of unrelated males cooperate to herd individual females. Multiple first-order alliances cooperate in teams (second-order alliances) in the pursuit and defence of females, and multiple teams also work together (third-order alliances). Yet it remains unknown how dolphins classify these nested alliance relationships. We use 30 years of behavioural data combined with 40 contemporary sound playback experiments to 14 allied males, recording responses with drone-mounted video and a hydrophone array. We show that males form a first-person social concept of cooperative team membership at the second-order alliance level, independently of first-order alliance history and current relationship strength across all three alliance levels. Such associative concepts develop through experience and likely played an important role in the cooperative behaviour of early humans. These results provide evidence that cooperation-based concepts are not unique to humans, occurring in other animal societies with extensive cooperation between non-kin.

[1] School of Biological Sciences, University of Bristol, Bristol, UK. [2] School of Biological Sciences, University of Western Australia, Crawley, WA, Australia. [3] Biology Department, University of Massachusetts Dartmouth, North Dartmouth, MA, USA. [4] Evolutionary Genetics Group, Department of Anthropology, University of Zurich, Zurich, Switzerland. ✉email: stephanie.king@bristol.ac.uk

Humans use accrued social knowledge to classify individuals into meaningful groups, which can encompass sporting teams, social castes and political alliances. Indeed, the maintenance of cooperation in human societies depends, in part, on our ability to classify relationships according to their cooperative payoffs[1–3]. While many animals are capable of distinguishing between in-group and out-group based on simple rules of familiarity or spatial location[4,5], others are also capable of classifying individuals who live in the same group using accumulated social information[6]. In social birds and mammals, for example, individuals interact in stable and predictable ways, allowing them to be classified into groups according to matrilineal kinship[7] or linear rank order[8–12]. Some primate species are even able to classify individuals hierarchically, by both kinship and rank[13,14]. Here, we examine the classification of social relationships in a non-human species that inhabits an extensive, multi-level social network with high levels of cooperation between non-kin[15].

Three decades of research on Indo-Pacific bottlenose dolphins (*Tursiops aduncus*) in Shark Bay, Western Australia, have revealed a fission–fusion grouping pattern in an open social network where individuals reside in a continuous mosaic of overlapping home ranges[16–18]. In contrast to many social birds and other mammals, bottlenose dolphins do not form territories or closed social groups (group composition can change on a minute-by-minute or hourly basis) and there is no sex-biased dispersal (both sexes are philopatric)[16,17]. Individuals can, however, form stable social relationships, with female association patterns in Shark Bay influenced by range overlap and maternal and biparental relatedness[19], and males forming long-term multi-level alliances of typically unrelated individuals[17,20]. In this system, young males are not part of a closed social unit from which to choose alliance partners but instead develop long-term bonds with other males within this open social network[20]. This differs from most terrestrial mammals, which live in semi-closed, often territorial groups with one or more reproductive females, and where maturing males tend to disperse to other groups[17].

In Shark Bay's multi-level alliances, male dolphins engage in coordinated efforts to compete with rival alliances over access to females[21,22]. At the first level, males cooperate in pairs or trios (first-order alliances) to herd single oestrus females during events termed consortships[17,21]. Males also cooperate in teams of up to 14 males in stable second-order alliances to attack other alliances for access to females and to defend against such attacks[17,21]. First-order allies are chosen from within a male's second-order alliance, with first-order alliances varying in composition and stability, i.e. some males show clear preferences for particular individuals with whom to partner as first-order allies, while others will consort females with numerous males from within their second-order alliance in a given mating season[17,23]. There is, however, a significant positive correlation between first-order alliance stability and consortship rate[17,23]. While first-order alliance relationships may be relatively labile, the second-order alliance can remain stable for decades and is considered the core unit of male social organisation[16,17]. First-order alliances are defined based on functional behaviour, i.e. herding a female together. Second-order alliances are defined using both quantitative measures, i.e. hierarchical clustering analysis of association indices[24], and functional behaviour, i.e. cooperating in the attempted theft and defence of females[17,21]. Further, some second-order alliances also form third-order alliances, involving significant association preferences (quantified using permutation tests[24]) among two or more second-order alliances that may support each other in the capture and defence of females from other alliances[17,25]. Third-order relationships thereby increase the chance of having allies nearby[17].

The nested alliance levels mean dolphins need to keep track of many different relationships, both at the individual and alliance level, which may pose significant cognitive challenges[15,21]. Indeed, bottlenose dolphins exhibit many of the cognitive skills humans possess to facilitate the monitoring of individual behaviour, including individual vocal labels similar to human names (signature whistles[26]), and social memory that spans decades[27]. Signature whistles are learnt, arbitrary signals that are structurally different between conspecifics, with identity information encoded in the frequency modulation pattern of the whistles independently of general voice features[28,29]. Signature whistles are stable over an individual's lifetime[29], and playback experiments have confirmed that dolphins use them to discriminate between individuals[28,30]. Our recent research illustrated that male dolphins in Shark Bay produce individually distinct signature whistles that do not bear similarities at the level of the second-order alliance[31]. So, if signature whistles cannot be used as a defining feature of membership in a second-order alliance, what information do male dolphins use to classify their relationships?

Some birds and mammals use common associations and transitive inference to classify group members based on kinship[7,32] and social bond strength[33–35], and similar traits should be readily available to the Shark Bay dolphins for associative learning. However, in the dolphins' social network, males are generally unrelated[20], and while second-order alliances can remain stable for decades, social bond strengths within the alliances are highly differentiated (Figs. 1 and 2 and Supplementary Fig. 1). Furthermore, bond strengths between some third-order allies are comparable to those between members of the same second-order alliance (e.g. Figs. 1 and 2), and socio-positive behaviours, such as petting (akin to mutual grooming in terrestrial mammals or preening in birds), occur across all three alliance levels. Second-order alliances can, however, be defined by the cooperative associations between individuals (i.e. support in the capture and defence of females). Signature whistles may thereby facilitate the recognition of second-order alliance or 'team' members if males learn the signature whistles of all those in their alliance network and use memory of previous cooperative interactions with individuals to classify their relationships.

We set out to test this team membership hypothesis using long-term (30 years) behavioural data combined with recent acoustic playback experiments and drone-mounted video to determine how adult male bottlenose dolphins classify their nested alliance relationships. We conducted playbacks with three second-order alliances that share overlapping home ranges and were known third-order allies (see 'Methods'). The males ranged in age from 28 to 40 years, many of whom have known each other for ≥28 years; members of the three alliances were sighted together as dependent calves or juveniles, years before they formed their respective second-order alliances (Fig. 2). We utilised a within-subject experimental design, where male dolphins were subject to signature whistle playbacks of males from their second-order alliance and males from their third-order alliance. As first-order alliances vary in composition and thus cannot be quantified as a binary classification, we used long-term behavioural records to determine the number of consortships (a proxy for first-order alliance stability[17,23]; Fig. 3) in which pairs of males had partnered to test if males responded more strongly to preferred first-order alliance partners (from within their second-order alliance). A behavioural response was measured as any visually detectable change in orientation and approach to the sound source and/or a vocal response. The strength of responses was measured with the variables response duration (seconds), approach distance (metres) and the presence/absence of a vocal response.

We used recent data to quantify the contemporary (i.e. 2018–2019, when playback experiments were conducted) social

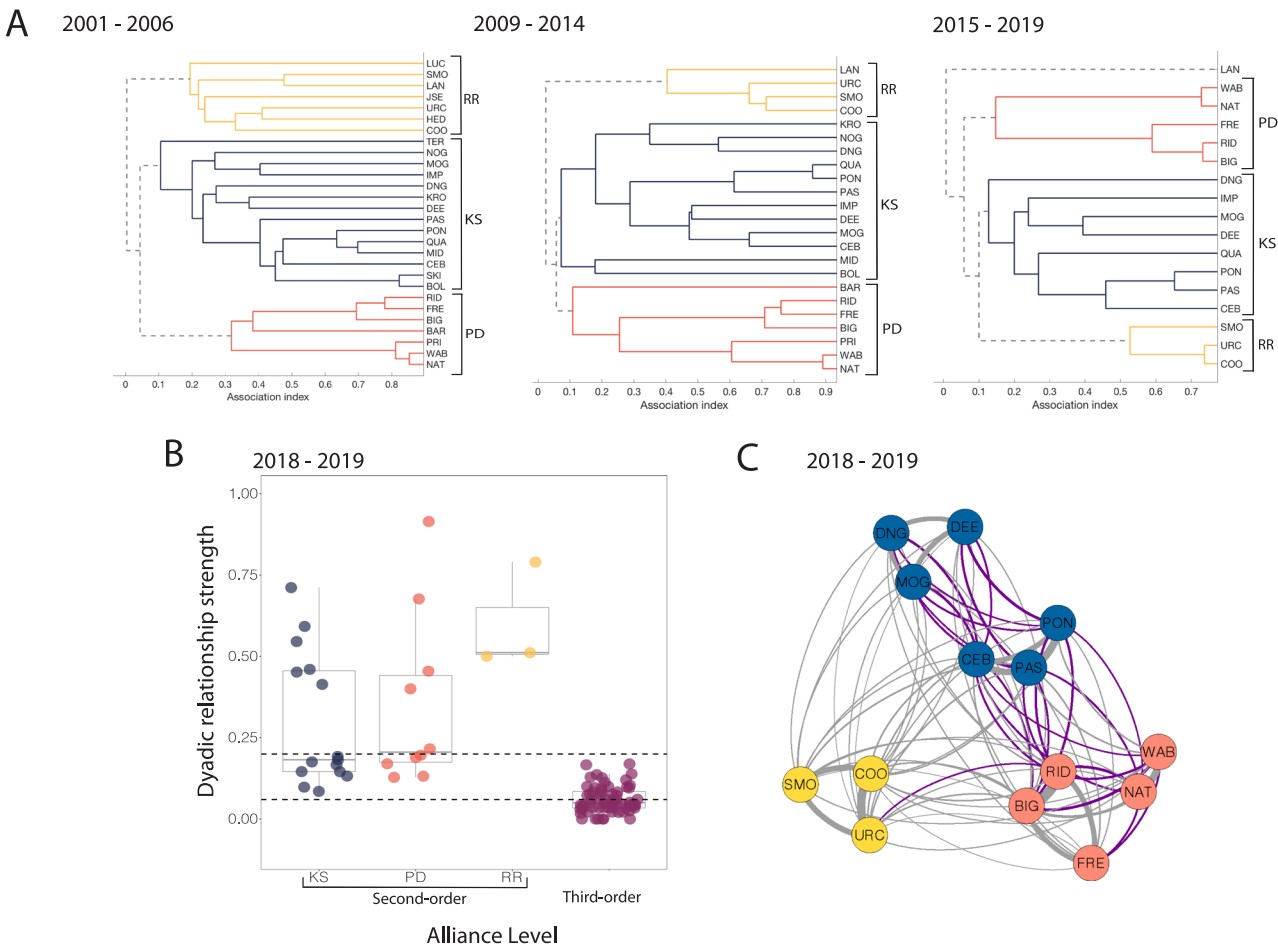

**Fig. 1 Multi-level alliance relationships. A** Average-linkage cluster diagram, based on association indices (Simple Ratio Index) among the three second-order alliances used in this study (KS = blue, PD = red, RR = yellow) over three time periods (2001–2006, 2009–2014, 2015–2019), demonstrating the stability of these second-order alliances. **B** Boxplots of association indices (also referred to as dyadic relationship strengths) during the playback study period (2018–2019), with actual values (coloured circles) for each second-order alliance (KS, PD, RR), as well as association indices between pairs of males in different second-order alliances (i.e. third-order allies). The solid grey horizontal lines represent medians, the boxes represent quartiles and the vertical lines represent the 1.5× interquartile ranges. The dashed horizontal lines delineate comparable bond strengths between male dyads within the same second-order alliance and male dyads at the third-order level. **C** Network plot of the three second-order alliances ($N = 14$ males) during the playback study period (2018–2019), with males colour-coded by alliance membership and purple edges representing comparable bond strengths between second- and third-order alliance levels (see **B**).

bond strength between the male whose signature whistle we played and the playback subject. Thus, we measured whether the strength of a male's response to his allies' signature whistles was driven by their current social bond strength, their consortship history (i.e. first-order partner level) or their second- or third-order alliance membership. If males respond more strongly to allies with whom they spend more time, then current social bond strength should predict their response strength. If males respond more strongly based on their history of first-order alliance cooperation, then consortship history should predict their response strength. If males use memory of previous cooperative second-order alliance interactions (i.e. assistance in the theft and defence of females from rival alliances) to classify relationships, then we would expect them to show the strongest response to playbacks of their second-order allies, irrespective of their differentiated relationships within those alliances (i.e. current bond strength or consortship history). However, if males classify all alliance relationships as being equivalent, then they should respond similarly to both second- and third-order alliance members.

In this study, we show that male dolphins classify second-order allies as members of their team, based on a shared cooperative

history. Males did not show a stronger response to first-order allies but responded strongly to all members of their second-order alliance, even those with whom they did not share a strong bond. Males did not respond as strongly to third-order allies, even if their social bond strength with the third-order ally played back was comparable to those with some second-order allies. This suggests that dolphins form a social concept of cooperative team membership at the second-order alliance level. The ability to classify relationships according to cooperative payoffs likely explains the prevalence of cooperation between unrelated individuals in human societies. Our findings reveal that cooperation-based concepts are not unique to humans but occur in other animal societies with extensive cooperation between non-kin.

## Results

We conducted 40 playbacks to 14 males, where each male was subject to a mean of 4.7 playbacks (range 1–8). The majority of playbacks were conducted to males when they were on their own (65%; 13 for second-order and 12 for third-order), with smaller proportions in pairs or trios with a female (20%; 3 for second-

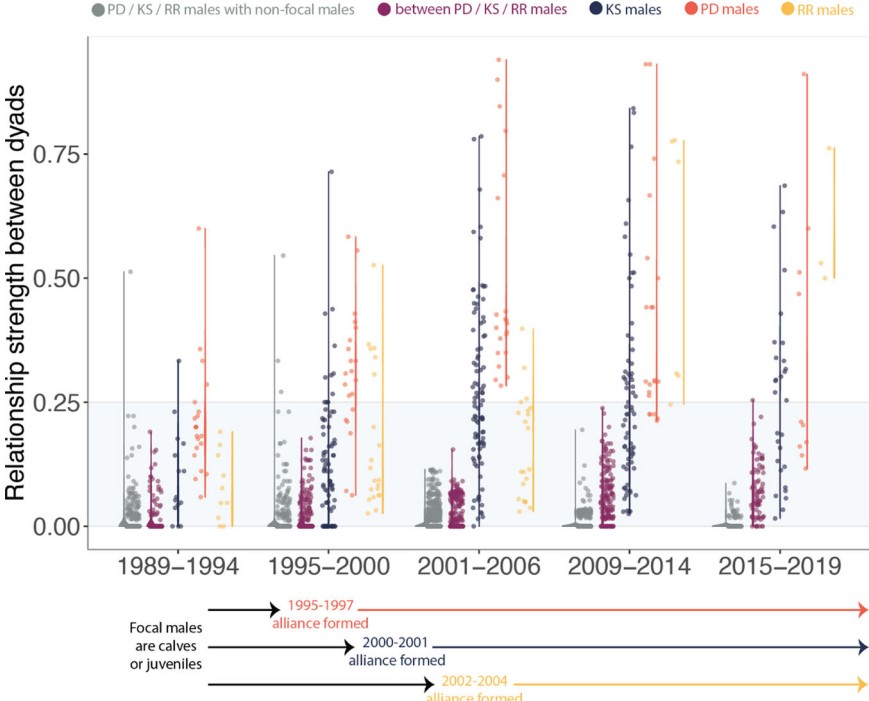

**Fig. 2 Dyadic relationship strength of males from three second-order alliances over a 30-year period.** Association indices (also referred to as dyadic relationship strengths) of males within three second-order alliances used in this study (KS = blue, PD = red, RR = yellow) calculated in 6-year snapshots over a 30-year period. Association indices between males in different second-order alliances (i.e. between PD, KS and RR males that later formed a third-order alliance) are shown in purple. Association indices between males in each of the three second-order alliances (PD, KS, RR) and other adult males in the population are shown in grey. The range of association values in each category is represented by the vertical line. The time period during which each second-order alliance formed (became sexually mature and started consorting females) is shown underneath the plot. The shaded area delineates comparable bond strengths between male dyads within the same second-order alliance and male dyads at the third-order alliance level (as well as those between focal and non-focal males). As the second-order alliances matured, some bond strengths increased but overall bond strength remained highly differentiated within each second-order alliance. Note: If males disappeared, we only used the years up until they were last seen to calculate their respective bond strengths with other males (i.e. we did not use the full 6-year period for those males).

| | KS alliance | | | | | | PD alliance | | | | | RR alliance | | |
|---|---|---|---|---|---|---|---|---|---|---|---|---|---|---|
| | PON (72) | PAS (63) | CEB (69) | DEE (80) | MOG (72) | DNG (48) | BIG (75) | FRE (69) | RID (71) | NAT (60) | WAB (60) | COO (62) | SMO (54) | URC (60) |
| PON | | | | | | | | | | | | | | |
| PAS | 33 | | | | | | | | | | | | | |
| CEB | 9 | 16 | | | | | | | | | | | | |
| DEE | 4 | 17 | 25 | | | | | | | | | | | |
| MOG | 6 | 3 | 33 | 25 | | | | | | | | | | |
| DNG | 1 | 6 | 3 | 13 | 12 | | | | | | | | | |
| BIG | 0 | 0 | 0 | 0 | 0 | 0 | | | | | | | | |
| FRE | 0 | 0 | 0 | 0 | 0 | 0 | 67 | | | | | | | |
| RID | 0 | 0 | 0 | 0 | 0 | 0 | 70 | 66 | | | | | | |
| NAT | 0 | 0 | 0 | 0 | 0 | 0 | 2 | 0 | 0 | | | | | |
| WAB | 0 | 0 | 0 | 0 | 0 | 0 | 2 | 0 | 0 | 60 | | | | |
| COO | 0 | 0 | 0 | 0 | 0 | 0 | 0 | 0 | 0 | 0 | 0 | | | |
| SMO | 0 | 0 | 0 | 0 | 0 | 0 | 0 | 0 | 0 | 0 | 0 | 40 | | |
| URC | 0 | 0 | 0 | 0 | 0 | 0 | 0 | 0 | 0 | 0 | 0 | 41 | 37 | |

**Fig. 3 Number of consortships in which each pair of males have partnered together.** Members of the KS alliance are shown in blue (# consortships since 2001), PD alliance in red (# consortships since 1996) and RR alliance in yellow (# consortships since 2002). Each second-order alliance was formerly greater in number (maximum alliance size of KS was 14 individuals, PD was 7 and RR was 7) but has subsequently diminished through member attrition. The total number of consortships recorded for each male is shown in parentheses.

order and 6 for third-order), or pairs or trios without a female (15%; 4 for second-order and 2 for third-order). In our analyses of playbacks to pairs and trios, we used either the bond strength and number of consortships of the male that showed the strongest response (which was always the male that responded first) or the maximum values if response strengths were the same. Both subject ID and caller ID were included as random effects. A visual and/or acoustic response was documented in 100% (20/20) of the second-order alliance playbacks we conducted and 75% (15/20) of

the third-order alliance playbacks. Individuals that responded to the playback stimuli generally did so immediately, with time to response averaging 2.5 s from the start of the signature whistle playback, and no detectable difference in latency between the two playback types (second- vs. third-order allies; Welch's $t$ test: $t = 0$, $N = 35$, $p = 1$; note, this test does not account for repeated measures). The strength of responses, however, was significantly different between the two playback types: individuals responding to playbacks of members of their second-order alliance exhibited

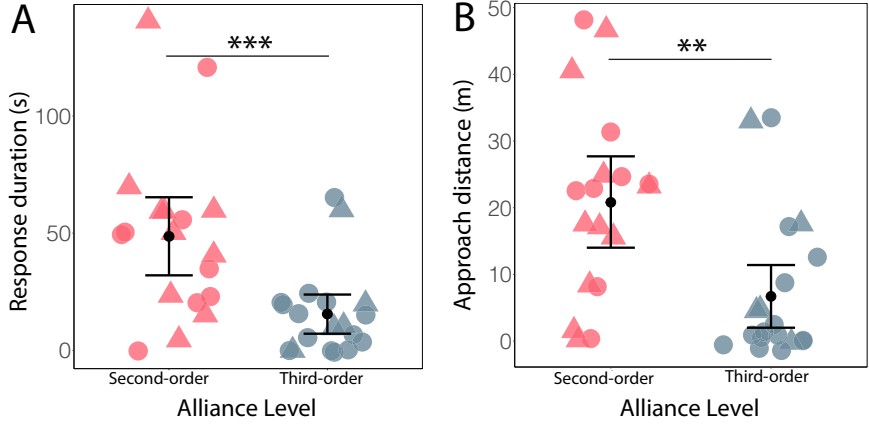

**Fig. 4 Behavioural response to alliance playbacks. A** Response duration ($N = 37$ playback experiments) and **B** approach distance ($N = 37$ playback experiments) by males to playbacks of second-order (red) and third-order (blue) allies. Black dots represent the mean, with 95% confidence intervals shown. Triangles represent playbacks for which the association indices (i.e. social bond strengths) between the male played back and the subject male(s) were between 0.1 and 0.2, i.e. relatively weak bonds for dyads within the same second-order alliance (red triangles), and relatively strong bonds for third-order alliance dyads (blue triangles) (see Fig. 1B). Asterisks denote statistical significance (***$p < 0.0001$; **$p = 0.002$; linear mixed-effect models). Source data are provided as a Source data file.

**Table 1 Parameter estimates for linear mixed-effect models for response duration ($N = 37$) and approach distance ($N = 37$) as a function of playback (PB) type (second- or third-order alliance), social bond strength (SRI), number of consortships (Consort) and PB order on a given day for a given subject.**

|  | Estimate | SE | Confidence interval | t value | p value |
|---|---|---|---|---|---|
| *(i) Response duration* | | | | | |
| Intercept | 9.3 | 8.5 | −7.4 to 26.9 | 1.0 | 0.2 |
| PB type (second order) | 60.3 | 13.4 | 33.4 to 86.0 | 4.4 | **<0.0001** |
| SRI | −20.4 | 40.7 | −82.3 to 63.0 | −0.5 | 0.6 |
| Consort | −0.8 | 0.5 | −1.8 to 0.04 | −1.5 | 0.1 |
| PB order | 16.9 | 9.7 | −1.5 to 35.5 | 1.7 | 0.09 |
| *(ii) Approach distance* | | | | | |
| Intercept | 4.2 | 3.9 | −3.1 to 11.7 | 1.0 | 0.2 |
| PB type (second order) | 20.5 | 6.3 | 8.5 to 32.2 | 3.2 | **0.002** |
| SRI | −6.5 | 19.0 | −41.8 to 29.5 | −0.3 | 0.7 |
| Consort | −0.1 | 0.2 | −0.59 to 0.31 | −0.5 | 0.5 |
| PB order | 6.1 | 5.0 | −3.3 to 15.6 | 1.2 | 0.2 |

Subject ID and caller ID were included as random effects. p values in bold indicate significant results. Baseline level is 'third-order' for PB type and 1 for PB order (the first or only playback on a given day).

significantly longer response durations ($p < 0.0001$, Fig. 4A and Table 1) and greater approach distances towards the playback source ($p = 0.002$, Fig. 4B and Table 1) than did individuals subject to playbacks of third-order allies.

Interestingly, neither current social bond strength nor the number of consortships predicted response strength. Thus, preferred first-order alliance partnerships did not drive response strengths (Table 1). Indeed, current social bond strength did not appear to influence response strength for either second- or third-order playback types (Fig. 4A, B). In total, we conducted 15 playbacks to the KS second-order alliance, 11 playbacks to the PD alliance and 14 playbacks to the RR alliance. To ensure the lack of differentiated relationships within the RR alliance (in which only a trio remain, Fig. 1) was not influencing the strong response to second-order alliance playbacks, we omitted the 14 RR alliance playbacks from a follow-up analysis. Even with the RR alliance removed, the results held; males exhibited significantly longer response durations (lmer estimate: 52.8, confidence interval (CI) = 20.3–85.3, $t = 2.9$, $p = 0.007$) and greater approach distances (lmer estimate: 26.6, CI = 8.2–36.4, $t = 3.5$, $p = 0.002$) to playbacks of members of their second-order alliance, and neither

current social bond strength nor number of consortships predicted response strength. Although we used total number of consortships to represent first-order alliance history, there was also a significant correlation between total number of consortships and the number of consortships recorded between dyads during this playback study (2018–2019, $N = 40$, $r = 0.98$, $p < 0.0001$).

Although it was not always possible to ensure that the subject(s) were moving directly away from the source (research vessel) prior to the playback, we were able to ascertain whether the animals turned towards the playback source when responding (Fig. 5). Subjects turned directly towards the playback source (i.e. irrespective of the magnitude of the change in orientation) in 90% (18/20) of the playbacks of second-order alliance members and 60% (12/20) of third-order ally playbacks (glmer estimate: 2.87, CI = 0.3–6.7, $z = 1.89$, $p = 0.05$; Supplementary Table 1). We recorded vocal responses (in which the subject male responded vocally to the playback) only twice, but both were to playbacks of second-order allies, and both involved the subject male replying with his own signature whistle (Supplementary Fig. 2). A description of the context associated with each playback is provided in the Source data file.

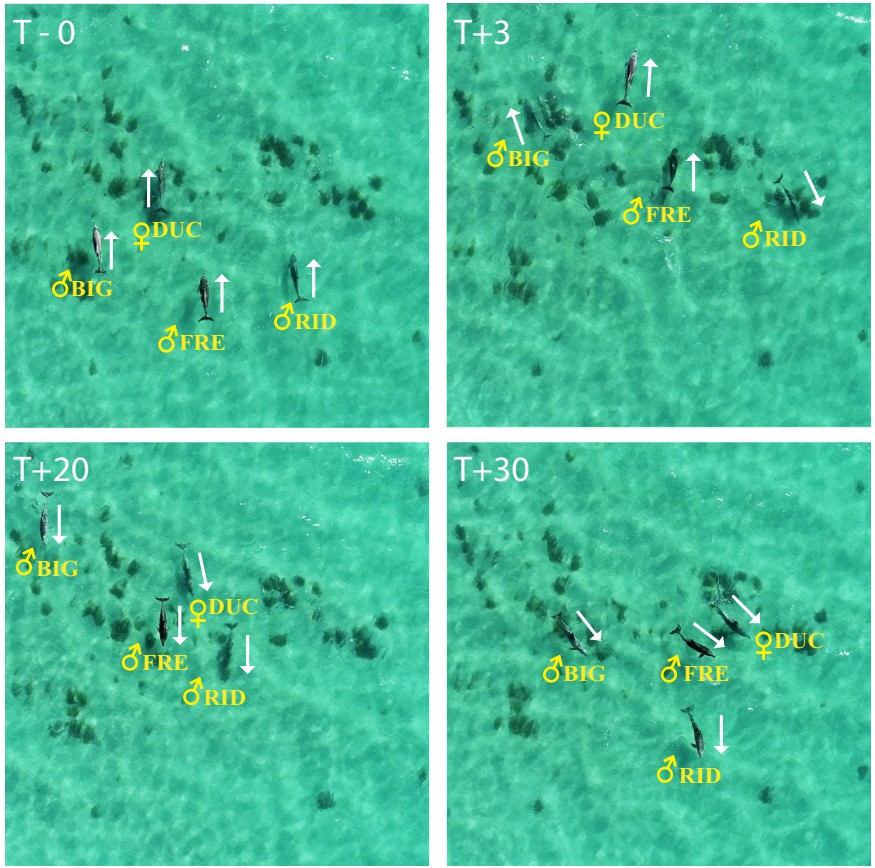

**Fig. 5 Behavioural response to second-order alliance playback.** Still images of a trio (first-order alliance BIG, FRE and RID) of male dolphins (♂) and their female consort (♀DUC) taken from drone-mounted video footage at the time of playback (T-0). Subsequent images at 3 s (T + 3), 20 s (T + 20) and 30 s (T + 30) after we played the signature whistle of their absent second-order ally (NAT). Animal orientation is identified by the arrows. Male RID turned immediately back towards the sound source (180°) and produced his signature whistle in response at T + 3 (see also Supplementary Fig. 2), after which, BIG, FRE and female DUC also turned (T + 20), before starting to veer away (T + 30). Source data are provided as a Source data file.

## Discussion

Effectively, our study used the highly differentiated relationships male dolphins have with their first- and third-order allies to demonstrate that they categorise second-order allies as members of a cooperative team. Males exhibited markedly stronger responses to second-order allies compared to third-order allies, even when social bond strength was comparable across those alliance levels. Further, they exhibited these responses independently of first-order alliance consortship history and contemporary relationship strength, i.e. male dolphins responded to second-order allies irrespective of first-order partner preferences. Strong responses were recorded by these males even to members of their second-order alliance with whom they had rarely or indeed never been documented partnering as first-order allies to consort a female. Males produce signature whistles that are notably different from members of their second-order alliance[31], they are not related[20] and the majority of the males in this study have been familiar with each other since they were calves or juveniles, years before they formed alliances (Fig. 2). Thus, neither whistle similarity, bond strength, kinship nor long-term familiarity can explain this result.

The male bottlenose dolphins' second-order team membership classification is likely based on the cooperative investment of individuals observed during the pursuit and defence of females or because the value of individual males to others may extend through the second-order alliance network. For example, while the number of consortships did not explain response strength, consortships do occur almost exclusively within the second-order

alliance. Even if male A has not consorted a female with male B, the triadic interactions within second-order alliances mean that it is likely that A has consorted a female with another male (e.g. C) that, in turn, has consorted a female with B. Investing in, and responding to, all second-order alliance members equally may therefore result in significant by-product benefits[36]. With regard to team-level cooperation, when pairs or trios (first-order alliances) of male dolphins partner in herding females, they can be subject to attacks by rival alliances in attempts to steal the female[17,21]. In many cases, first-order alliances have retained the female because members of their second-order alliance came to their defence, despite them gaining no obvious benefit[21], with the frequent association of second-order allies during the mating season likely preventing further attacks[17]. Indeed, the risk of injury for allied males during physical altercations with rivals can be high, and we have documented overtly aggressive encounters, including one male having been maimed in a fight over a female (unpublished data). Thus, help provided to a first-order alliance by second-order allies can be explained by direct reciprocity, where second-order allies support each other with the expectation that this will be reciprocated in the future, and/or pseudo-reciprocity, owing to their mutually dependent relationships[36,37]. Social reciprocity is thought to be an important evolutionary driver of cooperation among non-kin[38,39], and recent studies have shown that reciprocity predicts the long-term stability of partnerships between unrelated individuals in both wire-tailed manakins[40] and vampire bats[41]. Second-order alliances are stable over decades, with lost individuals rarely replaced[17],

creating the conditions for investment in 'team' members based on the past experience of direct and indirect reciprocity[36,37,42]. Indeed, evidence suggests that a number of social animals can integrate memories of past events to inform current decision-making[43–45], with reputation formation promoting cooperative relationships[1,2].

Male dolphins engaging in behaviour that involves considerable risks could be facilitated by arginine vasopressin, a neuropeptide hormone that regulates male mammalian social behaviours, such as affiliation and aggression[46]. Vasopressin has been shown to increase willingness in humans to engage in mutually beneficial, risky cooperation by reducing aversion to social risk[47]. It is conceivable that risky cooperation among male dolphins is mediated by spikes in vasopressin or another neuropeptide hormone, oxytocin. In humans, synchronous behaviour leads to oxytocin release[48], which promotes trust and cooperation[49,50]. Second-order allies engage in high levels of synchrony in the presence of a consorted female[51,52], and a link between oxytocin and prosocial behaviours has been demonstrated in other, non-human mammalian species[53–55]. Such hormonal mechanisms could be far more important for long-term bond strength than simple association, and these affective bonding mechanisms may explain why second-order allies responded so strongly to one another in this study.

As our results do not reflect the strongly differentiated relationships within second-order alliances, they suggest that male dolphins form a concept of cooperative 'team' membership at the second-order alliance level based on their reciprocal relationships. The formation of concepts allows individuals to group objects or events into distinct classes and there is evidence of three types of concept learning in non-human animals[56]. The first is perceptual or similarity-based concept learning, which involves grouping stimuli that share some physical property into classes[57]. The second is relational concept learning, which is the ability to report an object as same or different, i.e. are two pictures the same or different, or which of these two shapes is larger[58,59]. The third is associative concept learning, where arbitrary stimuli are classed together based on their association with a common event or prior association with each other[56]. Associative concepts are not based on physical properties or inherent relation among members but develop through experience[60]. We suggest that male dolphins are using first-person associative concept learning to classify males in their second-order alliance, based on their experience with each individual's arbitrary signature whistle and previous cooperative investment in the team. It remains unknown whether dolphins represent their dyadic relationships within the second-order alliance in a privileged way, where individual relationships are distinguished from one another, or whether all relationships are considered equivalent and grouped together into a conceptual category. To date, most work on social concepts in non-human animals has focused on third-person relationships[7,8,13]. Future work should determine whether the first-person social concepts used by male dolphins have predictive power and males can classify third-parties (i.e. non-allies) into their respective second-order alliances[6,7]. It is important to note here that, while many foraging specialisations exist within the Shark Bay dolphin population[61,62], we have yet to observe dolphins foraging cooperatively[17]. Individuals typically disperse to forage, and males that are not foraging spend their time cooperating in the pursuit and defence of females[17]. Thus, other forms of dyadic cooperation cannot explain our results.

To date, few studies have utilised playback experiments in the wild to determine how social bond strength influences individual responses to the vocalisations of group mates, with individuals responding more strongly to those with whom they share a stronger social bond (e.g. dwarf mongooses[34] and crested macaques[35]).

Vampire bats have been shown to respond more strongly to contact calls of past food-sharing partners[63]; however, this may also be explained by bond strength, as strong social bonds appear to be a pre-requisite for food sharing[41,45]. While there are clear benefits to having stronger bonds in dolphin alliances, as first-order allies that spend more time together have higher consortship rates[17,23], bond strength does not influence the response of allied males to the contact calls of cooperative partners. In the context of our playbacks at least, team membership surpasses both social bond strength and consortship history. Successful fights against rival alliances allow males to keep their female or acquire one. Thus, supporting and receiving support from second-order allies significantly contribute to a male's lifetime reproductive success. Allied males are known to increase whistle production during fusion events[64] and whistles are known to be part of a greeting sequence in bottlenose dolphins[65]. Males will produce their signature whistle to initiate a join and this may alert second-order allies that their support is required. This likely explains the strong responses we observed to the whistle playbacks of second-order allies. However, the responses we recorded should be considered low cost, so it remains to be determined whether higher cost investments accord with bond strength within second-order alliances. While third-order alliances are no doubt important in providing additional support in the capture or defence of females from rival alliances on occasion, this support is less consistent and the relationship includes more conflict, as third-order alliances are also competing for access to these females[25]. It is therefore second-order allies who provide consistent, cooperative support against rivals. Future work should determine whether males respond selectively to second-order allies relative to all other groups or whether they discriminate between all alliance levels (i.e. third-order allies vs. non-allies.)

Bottlenose dolphins are unusual in their invention of arbitrary vocal labels for individual recognition[28,30], but these labels facilitate the monitoring of individual behaviour and tracking of individual relationships. Captive bottlenose dolphins are capable of using novel, arbitrary signals to refer to objects[66], reporting on objects even in their absence, i.e. displaced reference[67], and developing relational concepts[68], indicating that some dolphins, at least, possess the cognitive skills to develop abstract concepts. Our work with wild dolphins illustrates that males can classify relationships based on the cooperative investment of individuals. While hormonal mechanisms may play a role here, we propose that this classification is evidence of associative concept learning, where male dolphins classify team members based on experience with each male's arbitrary signature whistle and history of cooperative investment. Such associative concepts develop through experience[56] and are believed to have played an important role in the cooperative social behaviour of early humans[69,70]. Our results illustrate that, in dynamic social environments with extensive cooperation between non-kin, both dolphins and humans are capable of such cooperation-based concept formation.

## Methods

Playback experiments were conducted during August 2018 and 2019 in the eastern gulf of Shark Bay, Western Australia, where our research on Indo-Pacific bottlenose dolphins (T. aduncus) has been carried out on a near-annual, seasonal basis (typically austral winter and spring) since 1982[17,71]. Our subject males were from three second-order alliances; the KS alliance (6 males), the PD alliance (5 males in 2018, 4 males in 2019) and the RR alliance (3 males). The KS and PD alliance had long been third-order allies[25] and, in more recent years (2013-), the RR alliance had forged a third-order relationship with KS and PD. The quantification of alliance relationships is described below. These males ranged from 28 to 40 years of age, many of whom have known each other for 28 years or more; members of all three alliances were sighted together as dependent calves or juveniles in 1989 and the early 1990s (Fig. 2). Male associations in early life have been shown to predict alliance partnerships as adults[20]. The respective second-order alliances had crystallised by the mid-1990s to the early 2000s (PD alliance 1995-1997; KS alliance 2000-2001; RR alliance 2002-2004).

The study was carried out in compliance with the ethics policies of the University of Bristol (Animal Welfare and Ethics Review Body) and University of Western Australia (Animal Ethics Committee). Permits for the scientific use of animals were obtained from the Department of Biodiversity, Conservation and Attractions, Western Australia.

**Quantifying male relationships.** Survey data were used to calculate the strength of social relationships between pairs of males. A survey is a minimum 5-min snapshot of dolphin group composition (as defined by the 10 m chain rule, where each dolphin in the group is within 10 m of any other dolphin[72]) and behavioural activity. Survey data are collected annually as part of our long-term research programme, with behavioural survey data spanning 36 years. Only association data recorded in the first 5 min of a survey were used to ensure association measures were comparable across surveys. Resights, where the same group is encountered within 2 h, were excluded. We calculated association indices using the Simple Ratio Index (SRI)[73,74] in the R package *asnipe*[75], which is an estimate of the proportion of time two animals spend together (0 for pairs of animals never observed together; 1 for pairs always seen together). Given the high degree of fission–fusion dynamics in bottlenose dolphin societies, association indices are a measure of bond strength and reflect true social preferences, i.e. individuals have more choice of associates than those living in relatively stable social groups[76].

Association indices, calculated from survey data collected between 2001 and 2019, of 92 males with extensive home range overlap were used to confirm second-order alliance membership. We used average-linkage hierarchical clustering in SOCPROG[24] to delineate second-order alliance membership (Supplementary Fig. 1), with a cophenetic correlation coefficient (CCC) of 0.98. A CCC >0.8 indicates that the hierarchical model provides a good representation of the social network[77]. The stability of these alliances is shown in Fig. 1. The membership of these second-order alliances corresponds to what we observe in the field, i.e. males determined to be in the same second-order alliance through average-linkage hierarchical clustering are the same males we observe herding females together and defending them from rivals[16,17,20,23,25,31,78]. We then examined associations between these second-order alliances by testing for inter-alliance preferences (i.e. third-order allies) in SOCPROG[24]. Permutation tests for preferred associations were run and the number of permutations was increased until the *p* value stabilised (PD vs KS, 10,000 permutations, *p* = 0.0002; KS vs RR, 10,000 permutations, *p* = 0.0001; PD vs. RR, 10,000 permutations, *p* = 0). These third-order alliance associations have also been confirmed in a previous study[25].

We then used survey data from 1989 to 2019 to calculate association indices (i.e. bond strength) for our focal males (KS alliance, PD alliance and RR alliance) in 6-year time periods from when they were calves and juveniles up to the present day (Fig. 2). We calculated association indices for focal males within the same second-order alliance, between focal males in different second-order alliances and between focal males and non-focal males (Fig. 2). We included all males that were originally part of the three second-order alliances (Fig. 1); when males disappeared, we only used the years up until they were last seen to calculate their respective association indices with other males (i.e. we did not use the full 6-year period for those males). We used survey data collected in 2018 and 2019 to calculate contemporary (i.e. when the playback experiments were conducted) association indices (i.e. social bond strength) between pairs of males across the three focal second-order alliances. For the 2018–2019 survey data, we had an average of 30 sightings (range: 15–41) for each of our 14 males.

Finally, we used our long-term data to tally the number of times each pair of males across all three second-order alliances had consorted a female together. Members of the PD alliance were first recorded consorting females in 1996, the KS alliance in 2001 and members of the RR alliance were first recorded consorting females in 2002. The number of consortships in which members of each alliance have participated varies considerably. Cases where members of different second-order alliances consorted a female together are rare, and none were observed during this study (Fig. 3).

**Playback stimuli.** To identify the signature whistle of each individual male (across the KS, PD and RR alliances), whistles were grouped together based on their frequency modulation patterns using an automated adaptive resonance theory neural network that incorporates dynamic time warping[79]. Whistles were then confirmed as signature whistles using the SIGID method[80], fully described as applied to these alliances in King et al.[31]. These signature whistles were used as stimuli for the playback experiments conducted here, each consisting of a signature whistle played twice, separated by a 2-s inter-whistle interval. We used Adobe Audition (version 12.1.0) to apply a high-pass filter at 2 kHz to each playback, and then standardised for amplitude (total root mean square (RMS)) across all stimuli by using the 'match loudness' function (target loudness −26 dB). While each playback consisted of the same signature whistle, we used different signature whistles across playbacks to minimise pseudoreplication, resulting in 27 unique playback stimuli across 40 playbacks ($\overline{x}$ number of different signature whistles used per male: 2, range: 1–3). Stimulus duration varied between signature whistle types but playbacks of second- vs. third-order allies to the same subject were matched for approximate duration. We compared stimuli durations between second-order ($\overline{x}$: 1.3 s, range: 0.6–2.2 s) and third-order alliances ($\overline{x}$: 1.3 s, range: 0.6–2.5 s) with Welch's *t* test and detected no difference ($N = 40$, $t = 37.37$; $p = 0.70$).

**Playback experiments.** Field experiments were carried out only in optimal observational and recording conditions (i.e. no rain, light winds only, sea state ≤3). Dolphin groups were approached to within 30 m for the purposes of conducting standard photo-identification and behavioural surveys. When potential subject males for playback experiments were encountered, we completed the survey and commenced a focal animal(s) follow, placing the hydrophone array (see below) into the water and awaiting suitable conditions (i.e. group composition, visibility, relative position of individuals to each other and to the research vessel/playback source). We allowed the subject individual/s to move 100–150 m from the research vessel before conducting the playback, an Unmanned Aerial Vehicle (UAV) was launched, the engine was switched off immediately prior and the underwater speaker placed in the water (details below). We made every attempt to ensure the animals were moving directly away from the research vessel at the onset of the playbacks (never conducting a playback when the group was moving towards us), and we ensured that the male whose signature whistle we were playing back was not nearby. Playback order was randomised (second- vs. third-order alliance), with a minimum of 30 min between playbacks and a maximum of two playbacks conducted per subject individual(s) on a given day.

Sounds were played back through a Lubell LL916C-025 (PRO) underwater speaker and power amplifier set (Lubell Labs Inc., Columbus, OH, USA.; frequency response: 200 Hz–23 kHz (500 Hz–21 kHz + / − 10 dB)) connected to a TASCAM DR-44WL digital recorder. The playback source level was approximately 141 dB re 1 μPa at 1 m (RMS), measured with a calibrated SoundTrap STD 300 (Ocean Instruments, NZ; sampling rate of 96 kHz; cliplevel of 177 dB re 1 μPa and a flat frequency response (±3 dB) from 0.02 to 60 kHz).

The UAV (DJI Phantom 4 Pro+), with an integrated, gimbal-controlled camera (with a focal length of 8.8 mm and maximum resolution of 3840 × 2160 pixels and 23 frames/s), was used to obtain high-quality visual records of how individual dolphins responded to the playback experiments. During each playback, the UAV was flown at altitudes ($\overline{x}$: 34 m, range: 30–42 m, as determined by onboard sensors) to maximise our ability to detect behavioural changes but also maintain a relatively broad field of view. On playback of each stimulus, every effort was made to hold the UAV stationary above the animals in order to best assess behavioural responses.

Responses to playbacks, including any visually detectable changes in orientation and approach distances to the source, were estimated in the field and corroborated with Bushnell laser range finders (±1 m accuracy) or UAV-mounted GPS. The playback and any vocal responses by the subject individual(s) were recorded using a four-element hydrophone array, consisting of four HTI-96 MIN series (flat frequency response: 0.002–30 kHz ± 1 dB) towed at 1 m depth around our research vessel in a rectangular formation (ca. 2.3 × 3.5 m), allowing us to later localise calls to individual dolphins. Audio recordings were made onto a TASCAM DR-680 MKII multi-track recorder at a sampling rate of 96 kHz. A spoken track was used to note the bearing (where the boat's bow was 0°), distance (m) and identification of the subject animals at each surfacing before and after each playback. Localisation accuracy of the array was calculated using custom-written MATLAB routines to calculate two-dimensional averaged MINNA (minimum number of receiver array) localisations (as per Quick et al.[81], Wahlberg et al.[82] and Schulz et al.[83]). The array was calibrated using two different frequency-modulated dolphin whistles, each approximately 1.5 s in duration, with a frequency range of 4–20 kHz. Acoustic localisation accuracy for whistle directions ($n = 75$) were calculated as 76% within ±15° of the true location and 99% within ±30°.

**Video analysis.** Using QuickTime player (v10.5), we scored the following variables from the videos of each playback experiment: (i) time to response (seconds)—time from the start of the first whistle stimulus to any visible response by the subject animal(s) to the playback; (ii) response duration (seconds)—time from the subject animal(s) first response to the playback until returning to pre-playback behaviour; (iii) approach distance (metres)—distance from first response until returning to pre-playback behaviour, estimated using adult dolphin body lengths (2 m) and corroborated with measurements and notes taken in real time from the research vessel; (iv) delta orientation (degrees)—relative change in heading on response to playback, estimated in degrees; and (v) orient to source (yes/no)—a binary measure of whether the subject animal oriented directly towards the playback source.

One observer (S.J.A.) coded all 40 playback videos from the UAV footage, as well as the time-synchronised audio recordings from the hydrophones and voice notes (using Adobe Audition 13.0.3.60). A second observer (S.L.K.), blind to playback treatments and the first observer's scores, independently coded variables (i) through (iv) for all 40 playback videos: time to response, response duration, approach distance, and delta orientation. We then conducted an inter-observer reliability analysis using the intraclass correlation coefficient (ICC) for two-way models in the irr package in R 3.6.1 (R project for statistical computing; GNU project) and found strong agreement between observers (ICC = 0.976, $p$ = < 0.0001, CI = 0.967–0.983). Given this parity, we used the data scores by observer one in all further analyses and provide all playback videos and corresponding data (see Data Availability section).

**Statistical analysis.** All statistical procedures were conducted in R 3.6.1 (R Core Team (2020)). To determine how behavioural response varied across playbacks, we built linear mixed-effect models (lmer using lme4 package in R) for two behavioural response parameters. We were unable to measure all response

variables to all playbacks (e.g. response duration if the animal dived out of view), so we provide sample sizes for each model below. We built two linear mixed-effect models for which the dependent variable was either response duration ($n = 37$ out of a possible 40) or approach distance ($n = 37$ out of a possible 40). Two of the omitted playbacks were the same for both response duration and approach distance (both to the RR alliance), and the third omitted playback was different (one further one to the RR alliance for approach distance, and one to the PD alliance for response duration). For each model, our fixed effects were playback type as a binary response (whistle from third-order alliance = 0, whistle from second-order alliance = 1), social bond strength (SRI) between the individual we played back and the playback subject (if the playback subject consisted of more than one allied male, we used the SRI with the male that showed the strongest response (which was always the male that responded first) or the maximum SRI if response strengths were the same), the number of consortships in which the individual we played back and the subject had partnered together (if the playback subject consisted of more than one allied male, we used the number of consortships with the male that showed the strongest response or the maximum number of consortships if response strengths were the same) and playback order on a given day for a given subject. We found no difference in response strength for males subjected to playbacks on their own or in groups, i.e. in pairs or trios with or without a female (see Supplementary Information), we therefore did not include group size in our models. Finally, we included both subject ID and caller ID as random effects (to account for repeated measures). To determine whether playback type influenced whether a male orientated towards the sound source, we built a generalised linear mixed-effect model with binomial family (glmer using lme4 package in R), where the dependent variable was orientation towards the source as a binary variable (no = 0, yes = 1). Fixed effects and random effects were the same as in the previous models. We calculated the variance inflation factor for each predictor to check for collinearity and all were <3[84], thus all predictor variables were retained in the model. We employed a traditional hypothesis testing approach where we used anova (car package in R) to test whether the full model (containing all our fixed effects, including the fixed effect of interest, i.e. playback type) explained significantly more variance than the null model (containing our fixed effects but without the fixed effect of interest, i.e. playback type). Results of the full-null model comparisons can be found in Supplementary Table 1. We also checked whether the inclusion of an interaction between playback type and social bond strength in our full models explained significantly more variance. We found no support for retaining the interaction and it was subsequently dropped from our models. Visual assessment confirmed that the residuals for the linear mixed-effect models were normally distributed. We calculated Cook's distances of single observations using the influence.ME package in R and detected no influential cases (all values <1)[85].

**Reporting summary**. Further information on research design is available in the Nature Research Reporting Summary linked to this article.

## Data availability

The playback videos are available from the Dryad Digital Repository (https://doi.org/10.5061/dryad.rbnzs7hb4). Source data are provided with this paper.

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

## Acknowledgements

We thank RAC Monkey Mia Dolphin Resort, Monkey Mia Wildsights and the Department of Biodiversity, Conservation and Attractions (DBCA) Shark Bay Rangers for their continued support and assistance. We particularly thank Pernille Sørensen for her help during this field study and Kelly Jaakkola for valuable comments on the manuscript. This dolphin research was carried out on Guthaaguda, Malgana Sea Country, and we acknowledge the traditional owners of the region. S.L.K. was supported by The Branco Weiss Fellowship—Society in Science and by grants from the National Geographic Society (050R-17 and 59136R-19).

## Author contributions

S.L.K. conceived study; S.L.K., S.J.A., R.C.C. and M.K. collected data and acquired funding; S.J.A. and S.L.K. coded the video and audio files; S.L.K. conducted the analysis and drafted the manuscript; all authors edited manuscript and approved submission.

## Competing interests

The authors declare no competing interests.
