## [Peer Review File · Nature Communications]

REVIEWER COMMENTS

Reviewer #1 (Remarks to the Author):

In this manuscript the authors use playback experiments to test the hypothesis that dolphins form a concept of group membership. They recorded if the response of male bottlenose dolphins to the playback of “signature whistles” from their second and third order alliances is dependent on general group membership or rather on social bond strength and courtships in the past. Their data shows that males bottlenose dolphins have stronger reactions when they listen to the whistle of a second order ally than third-order ally, when controlling for all the other factors. They conclude that dolphins share with humans the ability to classify conspecifics using abstract cooperation-base concepts.

Overall, this is a well-presented study and the dolphins seems to be an ideal candidate for the research question. The tables and figures are very nicely done and are useful to understand the main results. However, there are major issues with the statistical approach and the interpretation of the results. The manuscript would also benefit from clarifying some aspects of the methods.

General comments:

INTRODUCTION

1. Some of the information in the introduction should be streamlined. The text is slightly repetitive when it comes to explain why dolphins are special in terms of their social system and vocalizations.
2. In line with the previous point, some information is irrelevant for the current study. For example, “whistles” are described in detail and presented as a very distinctive characteristic of bottlenose dolphins. However, while the whistles might be special in the sense that are learned, other animals also produce vocal signatures (e.g. chimpanzees (Levréro & Mathevon, 2013)). For this study, it is not relevant if whistles are learned, but only that the other members of the group can recognize the caller.
3. In the introduction it is mentioned that birds and mammals can classify group members. Here it would be appropriate to mention that also fishes have some understanding of social concepts (even if this can be based simply on familiarity). For example, cleaner fish can differentiate resident and non-resident clients (Bshary and Grutter, 2002; Salwiczek et al., 2012; Triki et al., 2019).
4. The different predictions of the study are nicely explained (lines 136-144), but they are not linked to any hypothesis derived from the literature. Only the “team membership hypothesis” is mentioned, but not explained in any way.
5. The paragraph with the reciprocity (lines 103-106) seems out of context. The authors mention that the help provided to a first-order alliance can be explained by direct reciprocity or pseudo reciprocity,

but they do not mention how this relates to the study's question or how these concepts are integrated in the experimental design.

METHODS:

6. 35% of the data was recorded with more than one individual around. Can this influence the response e.g. by social facilitation or reputation formation? In those cases, the target individual was the one with a stronger response, why not choose randomly?

7. In order to minimize pseudo replication, the authors try to avoid using whistles from the same male more than once. Nevertheless, they end up using some whistles from the same male more than once (and it is not clear if their role is the same in each playback, that is, always a second order alliance members, always as third order alliance members, or one of each). Therefore, pseudo replication it is still an issue. To account for that they should include the caller ID in the model as a random factor.

8. The authors state that they "made every attempt to ensure the animals were moving directly away from the research vessel..." (line 360). Were these attempts always successful? If not, it can have an influence on the distance that the dolphins approached after the playback.

9. It is quite difficult to understand how the authors determined who belongs to which second and third order alliance and how the current and long-term bonds are quantified. What I understood is that they analyzed association data (from 2001 to 2019) using average-linkage hierarchical clustering, and they obtained three clusters (the three second order alliances). They also mentioned that functional behavior (e.g. defense of females) was included (line 271), but it is not clear how this data was included to calculate the different variables. What is not clear for me is what exactly are those association data (the SRI between all the male dyads?) and if it includes only the number of times that the animals were seen in proximity, or also if they attack other alliances or defend themselves from such attacks together (as this seems to be the proxy for second order alliances).

10. The description of how they calculate the variables related to the relationships between the males (membership of a second or third alliance, consort history and strength of the social bonds) it obscure and difficult to follow.

11. They use the survey data from 2018 to 2019 to calculate the "current" (at the time of the experiment) social bond strength between all the males at the time of the experiment. This raises a few questions. First, which data exactly they use (only proximity or also the number of times they consort a female together? Did they also include when the dyad was together as part of a second order alliance?). Secondly, the authors mentioned that first order alliances are not stable, but they do not give details about how often they change (within weeks? Months? Years?). That would be important to interpret how relevant are these measures to grasp the real social bond strength between the dyads at the time of the experiment. Finally, they have only an average of 30 data points (range 15 -41) per subject, and they use those data points to calculate the strength of the social bonds. Are 30 data points collected

across one year enough to reveal meaningful results about the strength of the social bonds?

12. They use long term data (30 years) to quantify the number of times that the pairs consort a female together. If two or three males consort a female together, it is supposed to indicate that they are in a first order alliance. Taking into account that first order alliances are not stable, it would make sense to include in the model not the data from the last 30 years, but also the number of times that a dyad consorted a female together in the last year.

13. The choice of the AIC method to select the best model is not justified by the design. They presented a design with a hypothesis, predictors and control variables, so there is no reason to select the best model. A more parsimonious approach would be to fit the full model comprising all the variables and compare it with a null model lacking the variables of interest. The use of AIC is not adequate when the same data is used to select the best model and, at the same time, draw inferences on it. It can be considered multiple testing and inflate type I error (Lubke & Campbell, 2016).

DISCUSSION

14. In the discussion the results are presented only in terms of significance. However, in the light of the wide CI reported for the main effect (17.23-73.8) indicates that the model should be interpreted carefully.

15. The authors do not discuss the limitations of the study (such as points 7, 8, 9 and 15) and the influence that they might have on the results.

16. The discussion does not help to understand the results, and some concepts are very vaguely explained. It would be interesting to find some lines in the discussion integrating the present study with the literature presented in the introduction (e.g. playback experiments done in other species) and explaining why the findings of this study are different. Specifically, it is not clear in the discussion which are the cognitive abilities that might be behind the results.

17. The fact that the males respond stronger to members of their second order alliance than to members of their third order alliance is not surprising. What it is surprising that the social bonds do not affect the subject's reactions to the playback. One should expect that social bonds would play some role here. Otherwise, why then have these closer bonds? A discussion about why they do NOT react stronger to their close partner seems necessary, but it is absent in the manuscript.

18. The authors claim that familiarity cannot affect the results because most of the males in this study have been familiar with each other since they were calves or juveniles (line 243). That is partially true, but even if the dolphins know each other, it is reasonable to think that they will see more often some individuals. Even if the strength of the social bonds is included in the model (but see also point 11.), the interaction between the social bond of a given dyad and if they both belong to the same second order alliance might have an effect. However, the model does not include interaction effects, so this idea cannot be tested. Even if there is some variability between the strength of the social bonds (2018-2019), as an average they should be all familiar at the second order alliance, as the first order alliances are not stable, and they switch preferred partners over the years.

19. In line with point 18, the authors suggest that male bottlenose dolphins classify second-order allies through direct experience of each individuals' cooperative investment (lines 148, 149). If that is true, it is difficult to understand then why the consort history does not have an effect, as mate grading is considered also a form of cooperation (King et al., 2019). Therefore, an individual can have experience with other individual cooperating in the context of a second-third order alliance.

Minor comments:

- Line 40 states "...in societies with extensive cooperation between non-kin, the formation of cooperation-based concepts is not unique to humans". As a reader, it is easy to interpret this sentence as a conditional ("in societies with extensive cooperation...concepts are not unique to humans, but in societies without extensive cooperation, it is"). It would be easier to read if it is rephrased ("Cooperation-based concepts are not unique to humans, but are also present in other animal societies with extensive cooperation between non-kin").
- In line 46 they spoke about cooperative payoffs and they include citations of articles that speak about reputation formation. They never discuss the relevance of reputation formation (if there is any) for this study.
- It would be nice to have some more information about the context where bottlenose dolphins usually emit their "whistles".
- Line 203/204: Sentence is difficult ... 'the male played back...'? There should be a better way to phrase it.
- To compare the latency to respond to the calls, they use a Welch's t test. I am not familiar with this test, but my understanding is that it should be used for unpaired samples. In this case, they compare the response from the same individuals to playbacks from two different individuals (paired samples). Why not use a test for paired samples like the t-student?
- Line 278: "the cognitive skills to link vocal labels to specific individuals based on direct association, and then organize" BUT this does not really seem to be the case, since social bonds do not seem to play a

role...

- Line 345 they say “the results show that dolphins classify”. That includes a little bit of interpretation. “I would say that the results show a different pattern of response according to....”.

References

Bshary, R. & Grutter, A. S. (2002). Experimental evidence that partner choice is a driving force in the payoff distribution among cooperators or mutualists: the cleaner fish case. *Ecol. Lett.* 5, 130– 136.

King, S.L., Allen, S.J., Krützen, M. et al. (2019). Vocal behaviour of allied male dolphins during cooperative mate guarding. *Anim Cogn* 22, 991–1000. <https://doi.org/10.1007/s10071-019-01290-1>

Levréro, F., & Mathevon, N. (2013). Vocal signature in wild infant chimpanzees. *American Journal of Primatology*, 75(4), 324-332. <https://doi.org/10.1002/ajp.22108>

Lubke, G. H., & Campbell, I. (2016). Inference Based on the Best-Fitting Model can Contribute to the Replication Crisis: Assessing Model Selection Uncertainty Using a Bootstrap Approach. *Structural equation modeling : a multidisciplinary journal*, 23(4), 479-490.

<https://doi.org/10.1080/10705511.2016.1141355>

Salwiczek, L.H., Prétôt, L., Demarta, L., Proctor, D., Essler, J., Pinto, A.I., et al. (2012) Adult Cleaner Wrasse Outperform Capuchin Monkeys, Chimpanzees and Orang-utans in a Complex Foraging Task Derived from Cleaner – Client Reef Fish Cooperation. *PLoS ONE* 7(11): e49068.

<https://doi.org/10.1371/journal.pone.0049068>

Triki, Z., Wismer, S., Rey, O., Binning, S.A. Elena Levorato, E., & Redouan Bshary, R. (2019). Biological market effects predict cleaner fish strategic sophistication, *Behavioral Ecology*, 30(6), 1548–1557.

<https://doi.org/10.1093/beheco/arz111>

Reviewer #2 (Remarks to the Author):

This manuscript describes a playback study with wild male bottlenose dolphins to test whether they classified individuals according to social association level. Males responded more strongly to playbacks of signature whistles produced by members of their team (the second-order alliance level) compared to playbacks of signature whistles produced by members of the third-order alliance, irrespective of dyadic bond strength between the calling individual and the target(s). The authors conclude that the dolphins form an abstract social concept of ‘team membership’.

The study appears generally well executed (although I have some questions regarding the experimental trials) and the results are interesting. I am not won over by the conclusion, however. In particular, the authors need to clearly define their idea of what an “abstract concept” is, and how one would distinguish between an account that is based on concept formation vs. one that is based on observational learning of who one’s team members are. If they can clean up the issues I raised and

provide a more nuanced discussion, I think this can make a very fine contribution to the literature.

In my view, an abstract concept refers to something that is not physically tangible. Whether or not 'team membership' constitutes an abstract concept is not a trivial question. In the present case, team membership arises out of association among multiple individuals. Not every member of the team needs to have a high association rate with each other member, but each member has one or a few strong associations with some or multiple team members. So, if both researchers and the animals classify other animals as being team members based on being observed together, does this provide evidence for the formation of an abstract concept? A more parsimonious explanation is that they register who is hanging out with whom, and thus consider who "belongs" to their team without having an abstract concept of a "team".

If the authors want to make the case for concept formation, they need to delve much deeper into the debate about concepts and concept formation, and re-evaluate their results more critically. There are some passages in the text that suggest the authors are also struggling with the theoretical framework (I can't blame them). For instance: in line 39, they write "such associative concepts develop through experience" – so now what is an "associative concept"? Is an associative concept different from an abstract concept?

The final paragraph of the discussion requires more care and precision. Reporting that previous studies found concept formation under captive conditions is not sufficient to infer concept formation in this particular case. The authors might want to discuss the points for and against such a conclusion. Having said this, I still think the results are interesting even if we cannot draw strong conclusions regarding the animals' mental representation of the social levels.

It would also be interesting to know what they make of the finding that Guinea baboon males responded more strongly to calls of their second-order alliance level than their third-order alliance level (Maciej et al. 2013. *Social monitoring in a multilevel society*, *Behavioural Ecology and Sociobiology*), while geladas show less differentiated responses (Bergman 2010. *Experimental evidence for limited vocal recognition in a wild primate PRSB*).

Another issue is that the authors claim that the association index is not a good predictor of team membership, as some dyads within teams have SRIs between .1 and .2, and some dyads at the third level have similar SRIs. Yet, Table 1 indicates that consortships exclusively occur inside teams, so this appears to be a perfect predictor. Having either been in consort with a certain other male, or in consort with a male that has been in consort with that male creates a good basis for assignment to social levels, it appears.

Further, I found the treatment of the trials where pairs or trios were tested (instead of single males) unclear. The authors write: "we used either the bond strength and number of consortships of the male that showed the strongest response, or the maximum value if response strengths were the same". Please clarify. It would have been much cleaner to predetermine whose response was going to be

measured prior to the analysis. It seems likely that you are introducing a bias here by selecting the strongest response.

Also, I was wondering if pairs or trios were evenly distributed across the two conditions – to check this piece of information and also more generally, the data should have been made accessible for the reviewing process already. This step in the analysis is really hard to understand.

A table with the number and order of trials per male should be included. Were all males tested twice on the same day? Order should be included in the models as a fixed factor.

Finally, I strongly recommend to omit the model selection part and simply do a full-null model comparison, with reporting of all the main factors in the full model if the full-null model comparison is significant.

Minor comments:

Line 81: do the authors insist on calling the signature whistles “arbitrary”?

84: change to “that the males in Shark Bay”

85: The authors should perhaps write: “... males produce individually distinct signature whistles that do not bear similarities at the level of the team, however”.

91: Please reconsider: “in the dolphins’ social network, there are no predictable traits that delineate second-order alliance membership, i.e. males are generally unrelated” – (i) these aspects are independent. Also “no predictable traits” is too general, as there is some trait (namely direct or indirect association) that allows for the delineation of the levels through cluster analyses.

95-114: could be deleted. The reader does not know where this is going and why this is relevant at this point. If the authors insist, some points may be integrated into the discussion.

173: Please check if the numbers add up: here you write that in 35 out of 40 playbacks (should be “playback trials”), the animals responded, but in line 181, $N = 34$. Why did you use a Welch’s t-test for unequal variances? Perhaps you could add a table in the supplementary material that clearly states which trial was used in which analysis, and which ones were omitted (see general comments).

180: add “in latency” before “between the two ...”

188/ Figure 2: open circles are very hard to read. Since a linear mixed model was used, it may be more appropriate to show the mean and SD. But I have no strong opinion on this. Panel C could be omitted; it is not very informative.

203: I would prefer to see Table S1 in the main text. If there are space constraints, Fig. 3 could go to the supplement.

235: change to: “irrespective of their contemporary dyadic bond strength or consortship history”

247: “more abstract”  see above.

299: perhaps specify: “within 10 m of any other dolphin”

345: “Stimulus duration” or “The duration of the stimuli”

395: how many frames per s?

396: change to “whistle stimulus”

416: give proper reference

421: I did not understand this sentence: “Two of the three ...”

426: again: this choice creates a bias (most likely). Either you need to determine the male ID prior to the analysis (or now have somebody select the target male blind to their response), or use the first responder, as this is the one that is more likely to be unaffected by the other males’ behaviour.

The following references require revision (put genus/species names in italics): 12, 34, 49

Avoid citing papers that have not been published yet (or else: put them on a preprint server if you want to cite them).

Ref 17 has the funny “Science (80-.)” that Mendeley produces; you can get rid of this when you select “Science N.Y.” as the journal title or something like this (if I remember correctly). I have a new version of Mendeley that is even worse so I cannot look that up.

623: A Fisher’s exact test is not applicable here, as it requires independent data points. You could to a mixed model with binomial error structure.

643: perhaps write “null model (marked with *)”

Table S2 is not needed

Julia Fischer

Reviewer #3 (Remarks to the Author):

The authors investigate dolphin's knowledge of their social structure using an interesting signature-whistle playback approach. Specifically, they present male dolphins with playbacks of second-order and third-order allies, finding that subjects are more responsive to second-order than third-order allies. The authors state that measures of dyadic bond strength and consort history cannot explain this difference (although as mentioned below I still have some confusion about these results). Field-experiments are always challenging and I'm sure they must be even more difficult with marine mammals, so the authors should be commended for the immense effort required to generate these data. The results are very interesting and they suggest that dolphins are discriminating individuals that are among their second-order allies from those that are among their third-order allies. However, as currently presented, I am not convinced of the authors' interpretation that dolphins possess social concepts or represent the nested structure of their alliance network.

Interpretation:

The authors describe second-order alliances as the core unit of male social organization (e.g., lines 67-68). From this perspective, one might see the observed effects as similar to work in diverse taxa suggesting that many species can discriminate familiar groupmates from less familiar or outgroup individuals (especially since there is no control data to see whether dolphins discriminate third-order allies from non-allies, which might support the view that dolphins have knowledge of a more nested structure than just more familiar/less familiar). It is interesting, and certainly worth discussion, that this boundary (between second-order allies and others) drives responses more strongly than dyadic metrics of bond strength or consortship history. However, it is hard to say whether discrimination reflects differences in cooperative history or lower-level characteristics like familiarity or time spent together or time spent interacting (which presumably differ most substantially between second- and third-order alliances, if second-order alliances are indeed the core unit of male social organization). It also seems possible that other aspects of dyadic relationship history (e.g., other forms of cooperation beyond consortship) that are high among all second-order allies could account for stronger responses to second-order alliances (i.e., that dolphins respond to second-order allies in a particular way because they consistently have particular [e.g., cooperative] relationships with those individuals; however, if there is no evidence that dolphins are grouping or categorizing those second-order allies together, then the pattern of consistently responding to second-order allies would owe to the shared characteristics of dolphin's personal dyadic relationships with those allies and not to their conception of the second-order alliance as a social unit).

If we are able to exclude dyadic relationship history (which I am not sure we can but which the authors may be able to argue more strongly for), then we might be able to conclude that dolphins can group members of their core unit and distinguish them from those outside their core unit on the basis of some traits common to their unit (perhaps cooperation). This would be a very interesting finding indeed and might reflect something like a proto-concept. However, for the following two reasons I am still not sure

whether it would amount to a proper social concept.

One issue is that unlike most work in animals aiming to characterize social concepts (in which animals must classify third-party relationships), in this study, the subjects themselves are part of the conceptual unit under study. If instead the authors had shown that dolphins could distinguish various second-order alliances of which they are not a member (e.g., outgroup alliance A from outgroup alliance B), this would be much stronger evidence that they conceive of second-order alliances as conceptual social units. Similarly, finding that dolphins could distinguish second-order alliances of which they are not a member from the third-order alliances that those second-order alliances contribute to would suggest awareness of the nested nature of their social structure (especially with further evidence that dolphins distinguish those various alliances from non-alliances). However, since dolphins could produce the reported results by only discriminating members of their own second-order alliance from individuals outside of that alliance, they may just track characteristics of their ingroup rather than a more nuanced social system or anything concept-like.

A second point is that social concepts are generally thought to have predictive power. Conceiving of particular social relationships (mother-offspring, allies, etc) conceptually allows me to generate predictions about conspecifics' behavior as soon as I identify them as a member of a particular conceptual class. For example, researchers like Cheney and Seyfarth have presented a variety of manipulations that suggest that monkeys have expectations of social behavior that stem from such classification (e.g., monkeys respond more strongly to playbacks of rank reversals between matriline than rank reversals within matriline and more strongly to rank reversal within matriline than rank-consistent playbacks, indicating expectations about the likelihood and consequences of these different simulated social events). Because dolphins were discriminating individual signature whistles rather than more complex social information, it is not clear whether they have any expectation about how their second-order allies will behave relative to other individuals.

More generally, conceptual understanding is a prominent topic in psychology, with major debates concerning whether young children and nonhuman animals are capable of any comprehension of concepts. Some (e.g., Seyfarth and Cheney, 2015, whom the authors reference) have proposed that certain species, like primates, are capable of generating basic and implicit conceptual knowledge while others have argued that the capacity to generate concepts is what separates humans from other species (Penn et al., 2008, Darwin's mistake: Explaining the discontinuity between human and nonhuman minds. Behavioral and Brain Sciences). If the authors wish to use the word 'concept' they should clearly define the term based on past literature and better defend its use.

Analyses:

I also have some questions about the analyses. First, from the main text I had the impression that the authors were employing a traditional hypothesis testing approach in which a full model is generated with all factors necessary to test different hypotheses and account for necessary controls. However, the methods section makes clear that a model selection approach has been used instead. The authors

should make this more explicit in the main text. Also, as far as I am aware, when employing a model selection approach, one should report the direction and strength of the effects in the best fitting model(s) using various terms but should not report p-values (as all factors included within the best-fitting models are considered important in explaining variation in the dependent measure).

Second, with regard to random effects, does 'playback target' refer to the subject? Subject ID should definitely be included as a random effect and it should be made clearer that this was the case. I would also be inclined to include as additional random effects the ID of the individual whose call has been played back and the second-order alliance of the subject. I appreciate that the limited sample size likely prohibited this possibility (since doing so would have made the models overly complex). However, since it would be appropriate to include such random effects, the authors should explicitly state in the methods that they could not include them as a result of sample size limitations.

Related to the above points, it is hard to interpret statements such as the following (lines 203-204): "Interestingly, neither the current social bond strength nor the number of consortships in which the male played back and the target male(s) were partnered over time predicted response strength." Does this mean that those terms were not represented in the best fitting models or that they were in the best-fitting model but were not significant? This is consequential for interpreting the relationship between the various predictors relative to the analysis approach (i.e., how alternative explanations have been controlled for). In viewing the supplementary Model Selection Tables, I find the reported effect confusing (that bond strength and consortships do not predict responses to the experimental manipulation) since the factor for consortships is present in the best-fitting model for response duration and the second best-fitting model for approach distance.

The authors should also be clearer in the main text whether there was only one best-fitting model or multiple (which appears to be the case from the supplementary Model Selection Tables), and which factors were retained in the best-fitting model(s).

General Clarity:

A few details should be moved from the methods section into the main text to ensure that readers can clearly understand the general details of the study without having to consult the methods.

-First, the authors should clarify the manipulation itself. In line 124, I wasn't sure whether 'paired signature whistles of either their second- or third-order allies' meant (1) that dolphins participated in a within-subject design in which they heard a single signature whistle per trial from a member of their second-order alliance (condition 1) or third-order alliance (condition 2; i.e., paired conditions), or (2) whether dolphins heard two signature whistles of different individuals in close succession (i.e., paired whistles within a condition). As it turns out, checking the methods revealed that both of my inferences were wrong. Paired referred instead to back-to-back signature whistles from the same individual. However, even after reading the methods section, I am not clear on how many trials each subject experienced and whether the design was fully within-subject or a mixed within- and between-subject

data set.

-Second (e.g., lines 124-130), the authors should be explicit about what counts as a response. As far as I can tell, this was not described at all in the main text and so I could not clearly process the results that immediately followed.

In the introduction, the structure of male alliances is well explained but I think the reader would benefit from description of how various-order alliances relate to broader group structure for males and females. That is, for males, are second-order alliances the unit of a typical social group (e.g., with shared territory) but these groups sometimes engage cooperatively (in third-order alliances) or competitively with neighboring second-order alliance groups? At what level are females considered part of the social group, and how stable are their associations with one another and with various levels of alliances?

Other Comments:

Line 85: Henceforth does not seem to be appropriate for this sentence

Overall, I find the work very interesting and if the authors could strongly support the claims that dolphins exhibit social concepts or that they can represent the nested structure of their alliance system, then I think the paper would definitely warrant publication in Nature Communications. However, as much as I would love to see this, at present, I'm not sure those claims can be satisfactorily defended with the available data.

Because the findings only pertain to discrimination of second-order allies from others (who happen to be third-order allies) but no discrimination of any other levels (e.g., third-order allies from non-allies), the experiment cannot expose whether dolphins understand the nested structure of their alliance system.

Whether the data could reflect a basic social concept depends most fundamentally, in my view, on whether the authors can exclude all possible dyadic traits that might co-vary with one's second-order alliance grouping and could explain more responsiveness to individuals who happen to be in one's second-order alliance. I appreciate the authors' effort to control for two such dyadic factors but I'm afraid their exclusion is not sufficient to confirm that another such variable isn't responsible. And without some additional evidence (e.g., discriminating between multiple social units that the subject is not part of, or generating expectations on the basis of classification), the evidence is insufficient to assume that dolphins categorize or group members of their second-order alliance in their minds.

REVIEWER COMMENTS

Reviewer #1 (Remarks to the Author):

In this manuscript the authors use playback experiments to test the hypothesis that dolphins form a concept of group membership. They recorded if the response of male bottlenose dolphins to the playback of “signature whistles” from their second and third order alliances is dependent on general group membership or rather on social bond strength and courtships in the past. Their data shows that male bottlenose dolphins have stronger reactions when they listen to the whistle of a second order ally than third-order ally, when controlling for all the other factors. They conclude that dolphins share with humans the ability to classify conspecifics using abstract cooperation-base concepts. Overall, this is a well-presented study and the dolphins seems to be an ideal candidate for the research question. The tables and figures are very nicely done and are useful to understand the main results. However, there are major issues with the statistical approach and the interpretation of the results. The manuscript would also benefit from clarifying some aspects of the methods.

Author response: We thank Reviewer #1 for the generally positive appraisal of our manuscript and for the very useful comments. Based on the feedback from all three reviewers, we have modified our statistical approach to address these concerns. Our results remain the same and we hope all the reviewers find our revised analyses much improved. We address each of the points raised below.

General comments:

INTRODUCTION

1. Some of the information in the introduction should be streamlined. The text is slightly repetitive when it comes to explain why dolphins are special in terms of their social system and vocalizations.

Author response: We have removed some of the text from the introduction (i.e. on reciprocity) and it is now more streamlined. Based on some of the comments from the other two reviewers, however, we felt it was important to highlight the differences between the dolphin’s social system and those of terrestrial mammals (for example, no territories or no stable groups) – see lines 54-63. These aspects are important to understand when interpreting our results in the context of the bottlenose dolphin social system.

2. In line with the previous point, some information is irrelevant for the current study. For example, “whistles” are described in detail and presented as a very distinctive characteristic of bottlenose dolphins. However, while the whistles might be special in the sense that are learned, other animals also produce vocal signatures (e.g. chimpanzees (Levréro & Mathevon, 2013)). For this study, it is not relevant if whistles are learned, but only that the other members of the group can recognize the caller.

Author response: We have removed some material around signature whistles being particularly interesting but deem it important to retain the remaining text in order to highlight how these whistles are ‘arbitrary’ in structure. In most species, the structure of the identity signal falls within a species-specific template with slight modifications encoding individual identity (Boughman and Moss 2003). In dolphins, however, signature whistles are structurally different between individuals, they are learnt and not inherited, and there appears to be no common element shared between population members; they are arbitrary in form. The use of arbitrary stimuli is now even more relevant as we discuss associative concept learning as a plausible explanation for our results (in our revised discussion), associative concept learning being the classification of arbitrary stimuli (i.e., signature whistles). We would thus prefer to keep this text describing signature whistles.

3. In the introduction it is mentioned that birds and mammals can classify group members. Here it would be appropriate to mention that also fishes have some understanding of social concepts (even if this can be based simply on familiarity). For example, cleaner fish can differentiate resident and non-resident clients (Bshary and Grutter, 2002; Salwiczek et al., 2012; Triki et al., 2019).

Author response: We have now added a reference on client recognition by cleaner fish, but we felt Tebbich et al., 2002 in Animal Cognition was the most apt reference.

4. The different predictions of the study are nicely explained (lines 136-144), but they are not linked to any hypothesis derived from the literature. Only the “team membership hypothesis” is mentioned, but not explained in any way.

Author response: Yes, we carefully considered all factors that could influence how males respond to each other’s calls and used these to develop our hypotheses and predictions. In our introduction and discussion, we explain how our predictions compare to findings in other animals, i.e., bond strength driving response strength in mongooses and macaques and history of cooperation driving response in vampire bats (although this is also correlated with bond strength): we now discuss this in lines (319-324). We have also modified the text in the introduction to more clearly explain the team membership hypothesis (lines 104-109).

5. The paragraph with the reciprocity (lines 103-106) seems out of context. The authors mention that the help provided to a first-order alliance can be explained by direct reciprocity or pseudo reciprocity, but they do not mention how this relates to the study’s question or how these concepts are integrated in the experimental design.

Author response: We agree and have now moved this section to the discussion where we explain our results more fully in the context of reciprocity, pseudo-reciprocity and mutual dependence (lines 264-287).

METHODS:

6. 35% of the data was recorded with more than one individual around. Can this influence the response e.g. by F or reputation formation? In those cases, the target individual was the one with a stronger response, why not choose randomly?

Author response: For the 35% of playbacks where > 1 male was present, response strength was either the same for all males or the strongest response was exhibited by the male that also responded first. Females never responded first but instead appeared to track the male’s behaviour. Thus, social facilitation cannot explain this result. We have now added more information on this in lines 184 and 509 and in our submitted data file (ESM).

7. In order to minimize pseudo replication, the authors try to avoid using whistles from the same male more than once. Nevertheless, they end up using some whistles from the same male more than once (and it is not clear if their role is the same in each playback, that is, always a second order alliance members, always as third order alliance members, or one of each). Therefore, pseudo replication it is still an issue. To account for that they should include the caller ID in the model as a random factor.

Author response: Repeats of the same male were used in both second- and third-order playbacks. We have now included caller ID as a random effect; so, both subject ID and caller ID are included as random effects.

8. The authors state that they “made every attempt to ensure the animals were moving directly

away from the research vessel...” (line 360). Were these attempts always successful? If not, it can have an influence on the distance that the dolphins approached after the playback.

Author response: We ensured that the animals were always 100-150m distant from the research vessel at the onset of the playback. The greatest approach distance was 48m, so the animals never approached to the maximum possible distance. While we made every attempt to ensure the animals were moving directly away from the research vessel at the onset of the playbacks, they were occasionally orientated parallel to the boat at the onset of the playback. We did not conduct any playbacks when the group was moving towards us. We were thereby able to measure approach distance for each playback. While the magnitude of the change in orientation was not always comparable – i.e., animals oriented directly away and turning 180° to approach the sound source/research vessel versus parallel animals turning 90°, we were able to ascertain whether the animals turned toward the playback source when responding. We now better clarify this in lines 229-231 and lines 440-441.

9. It is quite difficult to understand how the authors determined who belongs to which second and third order alliance and how the current and long-term bonds are quantified. What I understood is that they analyzed association data (from 2001 to 2019) using average-linkage hierarchical clustering, and they obtained three clusters (the three second order alliances). They also mentioned that functional behavior (e.g. defense of females) was included (line 271), but it is not clear how this data was included to calculate the different variables. What is not clear for me is what exactly are those association data (the SRI between all the male dyads?) and if it includes only the number of times that the animals were seen in proximity, or also if they attack other alliances or defend themselves from such attacks together (as this seems to be the proxy for second order alliances).

Author response: A standard “survey” of a dolphin group involves a minimum five-min snapshot of dolphin group membership and behaviour, photo-identifying all individuals in the group. The group is defined by the 10 m chain rule: all animals within 10 m of another animal are considered associated (so, yes, in close proximity). These survey data are used to quantify rates of association – in this case, using the Simple Ratio Index (SRI) - an estimate of the proportion of time two animals spend together (0 for pairs of animals never observed together; 1 for pairs always seen together). Second- and third-order alliance memberships are defined by association indices, calculated from survey data. We have now markedly expanded our methods section to provide more clarity in this regard. Given the high degree of fission–fusion dynamics in bottlenose dolphin societies, association indices are a measure of bond strength and reflect true social preferences, i.e., individuals have more choice in their associates than do those living in relatively stable social groups (we have now added this to lines 375-378). The membership of these second-order alliances is as described in previous studies (refs 4-7, 20-22, 24-25) and corresponds to what we observe in the field, i.e., males determined to be in the same second-order alliance through average-linkage hierarchical clustering are the same males we observe assisting others in their second-order alliance in stealing and defending females from rival alliances (we have now added this to lines 385-388). First-order alliances are defined by functional behaviour, i.e., whether they have consorted individual females together or not, which is why we use consortships as a measure for first-order alliance membership in the model (lines 75-76 and 117-119). We hope this is now clearer in the revised manuscript.

10. The description of how they calculate the variables related to the relationships between the males (membership of a second or third alliance, consort history and strength of the social bonds) is obscure and difficult to follow

Author response: Yes, as above, we have tried to expand on this description to make it clearer (see also response to comment 9). We do note here, however, that this text is very similar to that which we use in most of our publications and Reviewer #3 stated “the structure of male alliances is well explained”, so we hope that the edits/additions we have

now made render the description easier to follow.

11. They use the survey data from 2018 to 2019 to calculate the “current” (at the time of the experiment) social bond strength between all the males at the time of the experiment. This raises a few questions. First, which data exactly they use (only proximity or also the number of times they consort a female together? Did they also include when the dyad was together as part of a second order alliance?). Secondly, the authors mentioned that first order alliances are not stable, but they do not give details about how often they change (within weeks? Months? Years?).

That would be important to interpret how relevant are these measures to grasp the real social bond strength between the dyads at the time of the experiment. Finally, they have only an average of 30 data points (range 15-41) per subject, and they use those data points to calculate the strength of the social bonds. Are 30 data points collected across one year enough to reveal meaningful results about the strength of the social bonds?

Author response: These questions are easily addressed and, with respect, we do not deem it necessary to revise our methods section further. Yes, we use all survey data to calculate the association indices, i.e., bond strength – so, any instance in which the dyad was seen together, irrespective of what they were doing. As per response 9: A “survey” is a minimum five-min observation of dolphin group composition and behavioural activity. For this study, pairwise coefficients of association were calculated using the SRI (an estimate of the proportion of time two animals spend together). We restrict our unit of analysis to just those animals present in the first five mins of the survey to ensure that association measures are comparable across all surveys throughout the long-term study. Thus, bond strength is based on the amount of time individuals spend together, and this method is long-established in quantifying cetacean and other animal social networks. Previous research has shown that association patterns reflect true social preferences—given the high degree of fission–fusion dynamics in bottlenose dolphin societies: individuals have more choice in their associates than those living in relatively stable social groups¹ – we have now added this to lines 375-378. Yes, survey data from 2018 and 2019 were used to calculate contemporary (at the time of the study) bond strengths, and how they have changed (or not) over the course of data collection is quantified and portrayed visually in Figure 1 and the new Figure 2.

We do not state that first-order alliances are not stable, rather we describe how first-order alliances vary in composition and stability, i.e., some males showing clear preferences for particular individuals with whom to partner as first-order allies, while others consort females with several males from within their second-order alliance. We have now clarified that first-order alliances can change within a mating season (lines 69-72).

Finally, yes, our sample size is certainly enough to provide robust estimates of bond strength. A minimum of 15 sightings is used to measure bond strength, as this is when association indices tend to stabilise (using permutations, they actually stabilise at 10 sightings, but the standard deviation is below 1 for 15 sightings). Please see previous work by our group^{2,3} and another⁴⁻⁶, using 15-30 sightings (in one paper, 11 sightings sufficed⁷).

1. Galezo, A. A., Foroughirad, V., Krzyszczyk, E., Frère, C. H. & Mann, J. Juvenile social dynamics reflect adult reproductive strategies in bottlenose dolphins. *Behav. Ecol.* **31**, 1159–1171 (2020).
2. King, S. L. *et al.* Bottlenose dolphins retain individual vocal labels in multi-level alliances. *Curr. Biol.* **28**, 1993–1999 (2018).
3. Gerber, L. *et al.* Affiliation history and age similarity predict alliance formation in adult male bottlenose dolphins. *Behav. Ecol.* **31**, 361–370 (2020).
4. Frere, C. H. *et al.* Social and genetic interactions drive fitness variation in a free-living dolphin population. *Proc. Natl. Acad. Sci.* **107**, 19949–19954 (2010).
5. Frère, C. H. *et al.* Home range overlap, matrilineal and biparental kinship drive female associations in bottlenose dolphins. *Anim. Behav.* **80**, 481–486 (2010).
6. Krzyszczyk, E., Patterson, E. M., Stanton, M. A. & Mann, J. The transition to independence:

sex differences in social and behavioural development of wild bottlenose dolphins. *Anim. Behav.* **129**, 43–59 (2017).

7. Mann, J., Stanton, M. A., Patterson, E. M., Bienenstock, E. J. & Singh, L. O. Social networks reveal cultural behaviour in tool-using dolphins. *Nat. Commun.* **3**, (2012).

12. They use long term data (30 years) to quantify the number of times that the pairs consort a female together. If two or three males consort a female together, it is supposed to indicate that they are in a first order alliance. Taking into account that first order alliances are not stable, it would make sense to include in the model not the data from the last 30 years, but also the number of times that a dyad consorted a female together in the last year.

Author response: A first-order alliance is a functional classification, based on males working together to consort individual females. Thus, first-order alliances are defined by consortships (not by any other metric or criteria). We now make this clearer in lines 75-76. While we used total number of consortships to represent first-order alliance history, there is a significant correlation between total number of consortships and the number of consortships recorded between dyads during this playback study (N = 40, r = 0.98, P < 0.0001). We have now added this to lines 224-227 and provide raw data in our submitted data file (ESM).

13. The choice of the AIC method to select the best model is not justified by the design. They presented a design with a hypothesis, predictors and control variables, so there is no reason to select the best model. A more parsimonious approach would be to fit the full model comprising all the variables and compare it with a null model lacking the variables of interest. The use of AIC is not adequate when the same data is used to select the best model and, at the same time, draw inferences on it. It can be considered multiple testing and inflate type I error (Lubke & Campbell, 2016).

Author response: We have followed the advice of all the reviewers and have modified our statistical analyses. We now compare the full model with the null model. In all cases, the full model explained significantly more variance than the null model (Table S1). While we now acknowledge that we employed a traditional hypothesis testing approach (lines 522-523), we also present backwards model selection tables in the ESM (Table S2) for those readers that are interested to know which of the fixed effects significantly added to the model.

DISCUSSION

14. In the discussion the results are presented only in terms of significance. However, in the light of the wide CI reported for the main effect (17.23-73.8) indicates that the model should be interpreted carefully.

Author response: We now present the table with model parameters and CIs in the main text (Table 2) and have modified Figure 3 to show the mean and standard errors, so that the reader can assess the significance of the results directly. Please note that the CI for the main effects have changed slightly after we included the additional random effect (caller ID) and fixed effect (PB order). We have also added considerably to the discussion around our interpretation and plausible explanations of the results.

15. The authors do not discuss the limitations of the study (such as points 7, 8, 9 and 15) and the influence that they might have on the results.

Author response: We have included caller ID as a random effect in our model, thereby addressing point 7. We have further clarified in the revised manuscript that playbacks were never conducted when the subject animal/s were moving towards the research vessel/sound source (addressing point 8), and we provided more detail in the revised manuscript to address point 9. This is point 15, so we trust the above addresses it.

16. The discussion does not help to understand the results, and some concepts are very vaguely explained. It would be interesting to find some lines in the discussion integrating the present study with the literature presented in the introduction (e.g. playback experiments done in other species) and explaining why the findings of this study are different. Specifically, it is not clear in the discussion which are the cognitive abilities that might be behind the results.

Author response: We have now moved some material from the introduction to the discussion, and further revised our discussion to more fully explain the cognitive abilities that might be underlying our results (i.e., concept learning in animals) in lines 300-317. We have discussed other studies that conducted playback experiments to determine how bond strength influences response in lines 319-324.

17. The fact that the males respond stronger to members of their second order alliance than to members of their third order alliance is not surprising. What it is surprising that the social bonds do not affect the subject's reactions to the playback. One should expect that social bonds would play some role here. Otherwise, why then have these closer bonds? A discussion about why they do NOT react stronger to their close partner seems necessary, but it is absent in the manuscript.

Author response: The differential response between second- and third-order alliances is particularly surprising for those cases where bond strengths are comparable between second- and third-order allies. Within second-order alliances, bond strength is correlated with consortship rate, so there is a benefit to having strong bonds. However, successful fights against rival alliances allow males to keep their female or acquire a new one. Thus, supporting, and receiving support from, second-order allies also significantly contributes to a male's lifetime reproductive success. We now discuss this further in lines 324-329.

18. The authors claim that familiarity cannot affect the results because most of the males in this study have been familiar with each other since they were calves or juveniles (line 243). That is partially true, but even if the dolphins know each other, it is reasonable to think that they will see more often some individuals. Even if the strength of the social bonds is included in the model (but see also point 11.), the interaction between the social bond of a given dyad and if they both belong to the same second order alliance might have an effect. However, the model does not include interaction effects, so this idea cannot be tested. Even if there is some variability between the strength of the social bonds (2018-2019), as an average they should be all familiar at the second order alliance, as the first order alliances are not stable, and they switch preferred partners over the years.

Author response: We have added an additional figure (Fig 2) that shows the complete dyadic history of our focal males since they were calves/young juveniles (from 1989 to 2019). Bond strength is not based on grooming networks (as is the case for some terrestrial mammals), it is a measure of how much time two individuals spend together (0 for pairs of animals never observed together; 1 for pairs always seen together). So, we are accounting for how often these males see each other. Our new figure shows that bond strength (i.e. amount of time spent together) has always been highly differentiated between males both within and between alliances. With regards to including bond strength and alliance as an interaction in the model; interaction effects occur when the effect of one variable depends on the value of another variable. Our data offer strong evidence that alliance membership does not depend on bond strength given how differentiated these bonds can be, which is why an interaction is not included in the model. Also, as per point 11, we do not state that first-order alliances are not stable, rather that they vary markedly in stability (lines 69-72) – some are stable, some are labile. This is evident in Table 1, with consortship distribution.

19. In line with point 18, the authors suggest that male bottlenose dolphins classify second-order allies through direct experience of each individuals' cooperative investment (lines 148, 149). If that is true, it is difficult to understand then why the consort history does not have an effect, as mate grading is considered also a form of cooperation (King et al., 2019). Therefore, an individual can have experience with other individual cooperating in the context of a second-third order alliance.

Author response: We now discuss how this classification could be based on the history of team level cooperation during the pursuit and defense of females (direct reciprocity), or because the value of individual males to others may extend through the second-order alliance network (lines 264-287). For example, while the number of consortships between dyads did not directly explain response strength, consortships do occur exclusively within the second-order alliance. Even if male A has not consorted a female with male B, the triadic interactions within second-order alliances mean that it is likely that A has consorted a female with another male (e.g. C) that, in turn, has consorted a female with B. Investing in, and responding to, all second-order alliance members equally may therefore result in significant by-product benefits. We also cite new studies that show social reciprocity predicts the long-term stability of partnerships between unrelated individuals and that it is considered an important evolutionary driver for cooperation among non-kin (lines 281-283). Thus, the male's reciprocal relationships in the defense of females can explain why they respond to all second-order alliance members. We now make this clearer in lines 283-312. Further, as we point out in the discussion, the responses we measured can be considered relatively low cost and future studies may reveal that males discriminate within second-order alliances during high-cost interactions.

Minor comments:

- Line 40 states "...in societies with extensive cooperation between non-kin, the formation of cooperation-based concepts is not unique to humans". As a reader, it is easy to interpret this sentence as a conditional ("in societies with extensive cooperation...concepts are not unique to humans, but in societies without extensive cooperation, it is"). It would be easier to read if it is rephrased ("Cooperation-based concepts are not unique to humans, but are also present in other animal societies with extensive cooperation between non-kin").

Author response: Agreed. Thank you for the suggestion, we have made this change.

- In line 46 they spoke about cooperative payoffs and they include citations of articles that speak about reputation formation. They never discuss the relevance of reputation formation (if there is any) for this study.

Author response: We now mention reputation formation in the discussion, where we discuss social reciprocity (lines 285-287).

- It would be nice to have some more information about the context where bottlenose dolphins usually emit their "whistles".

Author response: We have now added this to lines 329-333 in the discussion.

- Line 203/204: Sentence is difficult ... 'the male played back...'? There should be a better way to phrase it.

Author response: We have now removed this part of the sentence.

- To compare the latency to respond to the calls, they use a Welch's t test. I am not familiar with this test, but my understanding is that it should be used for unpaired samples. In this case, they compare the response from the same individuals to playbacks from two different individuals (paired samples). Why not use a test for paired samples like the t-student?

Author response: A Welch's t-test accounts for unequal variances. We cannot use a paired t-test as we do not have equal sample sizes. Although we attempted to subject each male to same number of playbacks, this was not always possible due to the challenges of conducting these experiments in the field. We have now modified our description of the experiments in lines 115-117 and 179-182. We also provide all our raw data in the ESM.

- Line 278: “the cognitive skills to link vocal labels to specific individuals based on direct association, and then organize” BUT this does not really seem to be the case, since social bonds do not seem to play a role...

Author response: Bond strength does not have to play a role here. We propose that males match each male’s signature whistle to the memory of their previous cooperative interactions with that individual (irrespective of bond strength). Bottlenose dolphins are known to have long-term social memory (ref 28, Bruck 2013), which we highlight in the introduction. We have now also modified our concluding paragraph to say: “Captive bottlenose dolphins are capable of using novel, arbitrary signals to refer to objects, reporting on objects even in their absence, i.e., displaced reference, and developing relational concepts, indicating that some dolphins, at least, possess the cognitive skills to develop abstract concepts. Our work with wild dolphins shows that males can classify relationships based on the cooperative investment of individuals. We propose that this classification is evidence of associative concept learning; where male dolphins classify team members based on experience with each male’s arbitrary signature whistle and history of cooperative investment.”

- Line 345 they say “the results show that dolphins classify”. That includes a little bit of interpretation. “I would say that the results show a different pattern of response according to....”.

Author response: ‘Classify’ is defined as the grouping of “people or things” into classes or categories based on shared qualities or characteristics. We show that dolphins’ group second-order allies together (as demonstrated by their strong responses to second-order allies irrespective of bond strength or consortship history) based on their history of cooperative investment. We would, thus, like to keep the term ‘classify’ as we deem it apt.

Reviewer #2 (Remarks to the Author):

This manuscript describes a playback study with wild male bottlenose dolphins to test whether they classified individuals according to social association level. Males responded more strongly to playbacks of signature whistles produced by members of their team (the second-order alliance level) compared to playbacks of signature whistles produced by members of the third-order alliance, irrespective of dyadic bond strength between the calling individual and the target(s). The authors conclude that the dolphins form an abstract social concept of ‘team membership’.

The study appears generally well executed (although I have some questions regarding the experimental trials) and the results are interesting. I am not won over by the conclusion, however. In particular, the authors need to clearly define their idea of what an “abstract concept” is, and how one would distinguish between an account that is based on concept formation vs. one that is based on observational learning of who one’s team members are. If they can clean up the issues I raised and provide a more nuanced discussion, I think this can make a very fine contribution to the literature.

Author response: We thank Reviewer #2 for the largely positive appraisal of our manuscript and for the useful comments. We hope they will find our revised manuscript much improved. We have made significant revisions to address the methodological concerns (see detailed responses below) and to provide a more detailed/informed discussion on concept learning in animals. We agree that the most parsimonious explanation is that males learn who is in their team through direct experience of cooperative interaction. This can be explained by associative concept learning, which differs from the other two types of concept learning in animals; perceptual and relational. We now discuss these three concepts in lines 300-312 and how our results support associative concept learning. To avoid confusion, we have removed the emphasis on “abstract concept” as this term has been used in conjunction with all three forms of concept learning. We only refer to abstract concepts in relation to the work with captive dolphins.

In my view, an abstract concept refers to something that is not physically tangible. Whether or not 'team membership' constitutes an abstract concept is not a trivial question. In the present case, team membership arises out of association among multiple individuals. Not every member of the team needs to have a high association rate with each other member, but each member has one or a few strong associations with some or multiple team members. So, if both researchers and the animals classify other animals as being team members based on being observed together, does this provide evidence for the formation of an abstract concept? A more parsimonious explanation is that they register who is hanging out with whom, and thus consider who "belongs" to their team without having an abstract concept of a "team".

Author response: We have removed reference to researchers and dolphins classifying team membership in the same way as this perhaps blurs our reasoning. Further, we now provide complete dyadic relationship history (from 1989 to 2019) for all our focal males (see new Figure 2) to further show that bond strength (i.e., who is hanging out with whom) is highly differentiated within and between alliances and, therefore, cannot explain our results. Some dyadic bond strengths between third-order allies (i.e., members of different second-order alliances) are comparable to those between members of the same second-order alliance, yet target males typically did not respond to third-order allies. From Figure 4 in the manuscript, for example, we see a very strong response from RID to a playback of second-order ally NAT, despite their SRI being only 0.19 (i.e., they spend an estimated 19% of their time associating). The same male (RID) has a similar bond strength (0.17, they hang out 17% of the time) with his third-order ally PON, to whom he did not respond nearly as strongly. This pattern is reflective of our broader results, and cannot be explained by dyadic bond strength, or a fairly simple rule of "we spend time together". We argue, instead, that this provides compelling evidence for the animals having a concept of "belonging to the team" based on knowledge of cooperative history and cooperation with other members of their "team", irrespective of bond strength. Registering which males cooperatively invest in the team can be used to infer second-order alliance membership: associative concept learning is where arbitrary stimuli (e.g. signature whistles) are classed together based on their association with a common event or prior association with each other. Associative concepts are not based on physical properties or inherent relation among members, but develop through experience (lines 308-309). Our results support the notion that male dolphins use associative concept learning to classify males in their second-order alliance, based on experience with each male's arbitrary signature whistle and previous cooperative investment in the team. This assignment to the team could be based on a history of team level cooperation during thefts of females or defense against thefts (reciprocity), or because the value of individual males to others may extend through the alliance network (e.g. consortships occur exclusively in the team – see also our response to Reviewer 2's point below). We now explain this more fully in lines (264-287). We also propose that hormonal mechanisms may be far more important for long-term bond strength than simple association, and may explain why second-order allies respond so strongly to one another (lines 289-298).

If the authors want to make the case for concept formation, they need to delve much deeper into the debate about concepts and concept formation, and re-evaluate their results more critically. There are some passages in the text that suggest the authors are also struggling with the theoretical framework (I can't blame them). For instance: in line 39, they write "such associative concepts develop through experience" – so now what is an "associative concept"? Is an associative concept different from an abstract concept?

Author response: We now explain what we mean by associative concepts (one of the three types of concept learning in animals) in lines 300-312. To avoid confusion, we have removed the emphasis on "abstract concept" as this term has been used in conjunction with all three forms of concept learning. We only refer to abstract concepts once - in relation to the work with captive dolphins.

The final paragraph of the discussion requires more care and precision. Reporting that previous studies found concept formation under captive conditions is not sufficient to infer concept formation in this particular case. The authors might want to discuss the points for and against such a conclusion. Having said this, I still think the results are interesting even if we cannot draw strong conclusions regarding the animals' mental representation of the social levels.

Author response: We have revised the final paragraph to highlight the following points: (i) the results from captive dolphins show that some dolphins, at least, possess the cognitive skills to develop abstract concepts; (ii) our work with wild dolphins shows that males can classify relationships based on the cooperative investment of individuals; and (iii) we propose that this classification is evidence of associative concept learning, where male dolphins classify team members based on experience with each male's arbitrary signature whistle and history of cooperative investment – see lines 340-351.

It would also be interesting to know what they make of the finding that Guinea baboon males responded more strongly to calls of their second-order alliance level than their third-order alliance level (Maciej et al. 2013. Social monitoring in a multilevel society, *Behavioural Ecology and Sociobiology*), while geladas show less differentiated responses (Bergman 2010. Experimental evidence for limited vocal recognition in a wild primate PRSB).

Author response: The Guinea baboon study is very interesting, but there are a number of important differences with regards to their social system, i.e., Guinea baboons in the same gang are permanently associated and engage in socio-positive behaviours, while those in neighbouring gangs only share social proximity. Dolphins are not permanently associated with alliance members but reside in an open social network with lots of mixing; they have stable relationships, but not stable groups (we have tried to clarify this in lines 54-63) and socio-positive behaviours, such as petting, occur across all three alliance levels. We have now made this clear in line 101-103. It was interesting that the Maciej et al. 2013 discussion mentioned that the strength of relationships did not predict response strength, but because it was referred to as unpublished data, we decided to speculate on what might be going on here. We were unable to find anything further published in more recent papers but would be very keen to read the work if it has.

Another issue is that the authors claim that the association index is not a good predictor of team membership, as some dyads within teams have SRIs between .1 and .2, and some dyads at the third level have similar SRIs. Yet, Table 1 indicates that consortships exclusively occur inside teams, so this appears to be a perfect predictor. Having either been in consort with a certain other male, or in consort with a male that has been in consort with that male creates a good basis for assignment to social levels, it appears.

Author response: This is a very good point. It is true that indirect consortship associations could be a near-perfect predictor (there are unusual cases where males consort with others outside of their second-order alliance), as could direct cooperative support in attempts to steal or defend females (as we previously discussed in the manuscript). Both these mechanisms are based on experience of cooperative contribution to the team or team members - we now expand on this in the discussion in lines 264-287.

Further, I found the treatment of the trials where pairs or trios were tested (instead of single males) unclear. The authors write: "we used either the bond strength and number of consortships of the male that showed the strongest response, or the maximum value if response strengths were the same". Please clarify. It would have been much cleaner to predetermine whose response was going to be measured prior to the analysis. It seems likely that you are introducing a bias here by selecting the strongest response.

Author response: Response strength was either the same for all males (for the 35% of playbacks in which > 1 male was present) or strongest from the male that also responded first. Thus, social facilitation cannot explain our results. We have added more information

on this in lines 184 and 509 and in our submitted data file (ESM). All of our playback videos are also being submitted as ESM for transparency.

Also, I was wondering if pairs or trios were evenly distributed across the two conditions – to check this piece of information and also more generally, the data should have been made accessible for the reviewing process already. This step in the analysis is really hard to understand.

Author response: We have now added this information in lines 179-182 and in our submitted data file (ESM). We did submit our data file with the original submission but this may not have been shared when we transferred our manuscript from a different Nature journal.

A table with the number and order of trials per male should be included. Were all males tested twice on the same day? Order should be included in the models as a fixed factor.

Author response: We have included this information in our submitted data file (column G). Of the 40 playbacks, only 26 were conducted to the same subject on the same day. We now include playback order as a fixed effect in the models.

Finally, I strongly recommend to omit the model selection part and simply do a full-null model comparison, with reporting of all the main factors in the full model if the full-null model comparison is significant.

Author response: We have followed the advice of all the reviewers and modified our statistical analyses accordingly. We now compare the full model with the null model. In all cases, the full model explained significantly more variance than the null model (Table S1). We report all the main factors in the full model in Table 2. While we now acknowledge that we employed a traditional hypothesis testing approach (lines 522-523), we also present backwards model selection tables in the ESM (Table S2) for those readers that are interested to know which of the fixed effects significantly added to the model.

Minor comments:

Line 81: do the authors insist on calling the signature whistles “arbitrary”?

Author response: Given that signature whistles are structurally different between individuals, they are learnt and not inherited, and there appears to be no common element shared between population members, we feel the use of the term arbitrary is appropriate. Now that we have expanded on concept learning in animals, the use of arbitrary stimuli is rendered even more relevant for our discussion of associative concept learning. We would therefore like to keep the term arbitrary when discussing signature whistles.

84: change to “that the males in Shark Bay”

Author response: We have made this change.

85: The authors should perhaps write: “... males produce individually distinct signature whistles that do not bear similarities at the level of the team, however”.

Author response: We have changed this sentence to “Our recent research illustrated that the males in Shark Bay produce individually distinct signature whistles that do not bear similarities at the level of the second-order alliance.”

91: Please reconsider: “in the dolphins’ social network, there are no predictable traits that delineate second-order alliance membership, i.e. males are generally unrelated” – (i) these aspects are independent. Also “no predictable traits” is too general, as there is some trait (namely direct or indirect association) that allows for the delineation of the levels through cluster analyses.

Author response: We have modified this sentence so that it now reads “However, in the dolphins’ social network, males are generally unrelated and, while second-order alliances can remain stable for decades, social bond strength within the alliances is highly differentiated”.

95-114: could be deleted. The reader does not know where this is going and why this is relevant at this point. If the authors insist, some points may be integrated into the discussion.

Author response: Yes, we have removed this from the introduction and have integrated parts into the discussion (lines 264-287).

173: Please check if the numbers add up: here you write that in 35 out of 40 playbacks (should be “playback trials”), the animals responded, but in line 181, $N = 34$. Why did you use a Welch’s t-test for unequal variances? Perhaps you could add a table in the supplementary material that clearly states which trial was used in which analysis, and which ones were omitted (see general comments).

Author response: All our analyses are conducted in R and, when one runs a t-test in R, it will automatically select a Welch’s t-test if the variances are unequal. Sample size was 34 because one response was acoustic only (not physically observable), however, it was still a response and a mistake to omit that trial. We have now included it in the t-test. We have provided a data file with all data used in our analyses, so our results can be replicated and it is clear which playbacks were used for which analyses.

180: add “in latency” before “between the two ...”

Author response: We have made this change.

188/ Figure 2: open circles are very hard to read. Since a linear mixed model was used, it may be more appropriate to show the mean and SD. But I have no strong opinion on this. Panel C could be omitted; it is not very informative.

Author response: We have switched to a bar plot showing the mean with standard error bars. We have kept the raw data in the figure but have modified it to make it easier to read (triangles instead of open circles). We have moved Panel C to the ESM as a separate figure.

203: I would prefer to see Table S1 in the main text. If there are space constraints, Fig. 3 could go to the supplement.

Author response: We have now moved Table S1 to the main text (now Table 2).

235: change to: “irrespective of their contemporary dyadic bond strength or consortship history”

Author response: We have modified this section to address this wording, so the change no longer applies (see lines 251-256).

247: “more abstract”  see above.

Author response: This has been removed.

299: perhaps specify: “within 10 m of any other dolphin”

Author response: We have made this change.

345: “Stimulus duration” or “The duration of the stimuli”

Author response: We have changed to ‘Stimulus duration’.

395: how many frames per s?

Author response: We clarified this in the “Playback experiments” section of the revised manuscript, stating “The UAV (DJI Phantom 4 Pro+), with an integrated, gimbal-controlled camera (with a focal length of 8.8 mm and maximum resolution of 3,840 × 2,160 pixels and 23 frames/s), was used to obtain high quality visual records of how individual dolphins responded to the playback experiments.” (now lines 452-453)

396: change to “whistle stimulus”

Author response: We have made this change.

416: give proper reference

Author response: We have now added this reference.

421: I did not understand this sentence: “Two of the three ...”

Author response: We have modified the wording slightly to make this clearer: “We built two linear mixed-effect models for which the dependent variable was either *response duration* (n = 37 out of possible 40) or *approach distance* (n = 37 out of possible 40). Two of the omitted playbacks were the same for both response duration and approach distance (both to the RR alliance), and the third omitted playback was different (one further one to the RR alliance for approach distance, and one to the PD alliance for response duration).”

426: again: this choice creates a bias (most likely). Either you need to determine the male ID prior to the analysis (or now have somebody select the target male blind to their response), or use the first responder, as this is the one that is more likely to be unaffected by the other males’ behaviour.

Author response: Response strength was either the same for all males (for the 35% in which > 1 male was present), or the strongest response was always exhibited by the male that also responded first. Thus, social facilitation cannot explain this result. We have added more information on this in lines 184, in the methods 509 and in our submitted data file (ESM).

The following references require revision (put genus/species names in italics): 12, 34, 49

Avoid citing papers that have not been published yet (or else: put them on a preprint server if you want to cite them).

Author response: We have checked our references again to ensure Genus and species are in italics where necessary and have removed the two unpublished papers.

Ref 17 has the funny “Science (80-.)” that Mendeley produces; you can get rid of this when you select “Science N.Y.” as the journal title or something like this (if I remember correctly). I have a new version of Mendeley that is even worse so I cannot look that up.

Author response: Thank you! This is a very useful tip.

623: A Fisher’s exact test is not applicable here, as it requires independent data points. You could to a mixed model with binomial error structure.

Author response: We have replaced this analysis with a mixed model with binomial error structure and have moved these results to the main text.

643: perhaps write “null model (marked with *)”

Author response: We have now removed the model selection tables and replaced them with the full-null model comparison (Table S1).

Table S2 is not needed

Author response: We have removed the old Table S2.

Reviewer #3 (Remarks to the Author):

The authors investigate dolphin’s knowledge of their social structure using an interesting signature-whistle playback approach. Specifically, they present male dolphins with playbacks of second-order and third-order allies, finding that subjects are more responsive to second-order than third-order allies. The authors state that measures of dyadic bond strength and consort history cannot explain this difference (although as mentioned below I still have some confusion about these results). Field-experiments are always challenging and I’m sure they must be even more difficult with marine mammals, so the authors should be commended for the immense effort required to generate these data. The results are very interesting and they suggest that dolphins are discriminating individuals that are among their second-order allies from those that are among their third-order allies. However, as currently presented, I am not convinced of the authors’ interpretation that dolphins possess social concepts or represent the nested structure of their alliance network.

Author response: Thank you for your thoughtful and constructive comments. We hope our detailed responses to your comments below has alleviated any confusion and convinced you of the significance of our results.

Interpretation:

The authors describe second-order alliances as the core unit of male social organization (e.g., lines 67-68). From this perspective, one might see the observed effects as similar to work in diverse taxa suggesting that many species can discriminate familiar groupmates from less familiar or outgroup individuals (especially since there is no control data to see whether dolphins discriminate third-order allies from non-allies, which might support the view that dolphins have knowledge of a more nested structure than just more familiar/less familiar). It is interesting, and certainly worth discussion, that this boundary (between second-order allies and others) drives responses more strongly than dyadic metrics of bond strength or consortship history. However, it is hard to say whether discrimination reflects differences in cooperative history or lower-level characteristics like familiarity or time spent together or time spent interacting (which presumably differ most substantially between second- and third-order alliances, if second-order alliances are indeed the core unit of male social organization).

Author response: We did indeed have to grapple with how best to convey the take-home message, including how to avoid having the results misconstrued as “in-group vs out-group”. Here, we respectfully disagree that our results can be explained by an ingroup/outgroup or more familiar/less familiar divide. This is primarily because males in third-order alliances are not less familiar with one another other than those in second-order alliances. All these males were born into the same geographical area, their mothers sharing broad home range overlap, and they socialised as calves and juveniles, years before the alliances were formed. It is also important to consider this in the context of the dolphin social system, which we now expand on in the introduction (lines 54-63). All these males reside in an open social network with high encounter rates and ample opportunity for mixing of individuals. These dolphins exhibit a fission-fusion grouping pattern and do not form stable social groups or establish territories, as occurs in many birds and terrestrial mammals (lines 54-63). In order to help demonstrate that all our focal males are familiar

with one another, we have added a new figure (Fig. 2) that shows dyadic association indices (i.e., time spent together) between these males from when they were calves and juveniles to the present day (1989 to 2019). The association index we use is a well-established measure of time spent together and Fig. 2 shows clearly that associations (i.e., time spent together) have always been highly differentiated both within and between alliances. Given the high degree of fission–fusion dynamics in bottlenose dolphin societies, association indices are a measure of bond strength and reflect true social preferences, i.e., individuals have more choice in their associates than those living in relatively stable social groups (we have now expanded our methods section to make this clearer 375-378). We also now make it clear that socio-positive behaviours, such as petting (akin to primate grooming) occurs across all three alliance levels (lines 100-103). As such, neither long-term familiarity nor time spent together can explain these results.

It also seems possible that other aspects of dyadic relationship history (e.g., other forms of cooperation beyond consortship) that are high among all second-order allies could account for stronger responses to second-order alliances (i.e., that dolphins respond to second-order allies in a particular way because they consistently have particular [e.g., cooperative] relationships with those individuals; however, if there is no evidence that dolphins are grouping or categorizing those second-order allies together, then the pattern of consistently responding to second-order allies would owe to the shared characteristics of dolphin's personal dyadic relationships with those allies and not to their conception of the second-order alliance as a social unit).

Author response: Unlike some of the pelagic species, Indo-Pacific bottlenose dolphins (and various other shallow water species) are not known to cooperate in contexts other than mate acquisition, e.g., foraging. Cooperative foraging is clearly not pervasive in Shark Bay; in fact, after more than 35 years of observation, we have not observed dolphins (males or females) foraging cooperatively. Dispersing to forage is widespread among all dolphins in the bay (ref 4, Connor and Krützen 2015 and lines 313-317). While all males form alliances, alliances may specialise in foraging in very different habitats (O'Brien et al. 2020 Anim Beh) and, for some, dispersed foraging predominates. Cooperative foraging thus cannot be used to explain our results. If these males aren't foraging or resting, then they are consorting females – that is all they do. We are aware of no other dyadic interaction from over 35 years of careful observation. We have accounted for their full dyadic history with regards to how much time the males spend together (new Figure 2) and their history of consorting females (Table 1). The new Figure 2 shows that bond strength (i.e., who is hanging out with whom) is highly differentiated within and between alliances and, therefore, cannot explain our results. But cooperation in the pursuit and defence of females occurs among all second-order alliance members - we now expand on this in the discussion in lines 264-287. Our hypotheses (lines 96-137) were carefully designed to address all aspects of these male's relationships and, again, it is important to place these results in the context of the Shark Bay dolphin social system (see comment above and lines 54-63).

If we are able to exclude dyadic relationship history (which I am not sure we can but which the authors may be able to argue more strongly for), then we might be able to conclude that dolphins can group members of their core unit and distinguish them from those outside their core unit on the basis of some traits common to their unit (perhaps cooperation). This would be a very interesting finding indeed and might reflect something like a proto-concept. However, for the following two reasons I am still not sure whether it would amount to a proper social concept.

Author response: Yes, we have now made some important changes to our manuscript that, we trust, address the concerns above. First, we provide a much clearer description of the dolphin social system within which these results should be considered (lines 54-63). Second, we provide the full, 30-year dataset on dyadic relationship history, i.e., measures of the time males have spent together since they were calves (new Figure 2). Third, we make it clear that cooperative foraging, if it exists at all, cannot explain our results (lines 313-317) and, if males are not foraging, they spend their time herding females (lines 315-316). Thus, the time these males spend together (i.e., bond strength), the number of consortships they

have participated in together, and their cooperative support in the defence of females are the three traits that can define these alliances. Our results illustrate clearly that bond strength and dyadic consortship history do not drive response strength. Instead, males distinguish second-order alliance members based on cooperative support in the defence of females - we now expand on this in the discussion in lines 264-338.

One issue is that unlike most work in animals aiming to characterize social concepts (in which animals must classify third-party relationships), in this study, the subjects themselves are part of the conceptual unit under study. If instead the authors had shown that dolphins could distinguish various second-order alliances of which they are not a member (e.g., outgroup alliance A from outgroup alliance B), this would be much stronger evidence that they conceive of second-order alliances as conceptual social units. Similarly, finding that dolphins could distinguish second-order alliances of which they are not a member from the third-order alliances that those second-order alliances contribute to would suggest awareness of the nested nature of their social structure (especially with further evidence that dolphins distinguish those various alliances from non-alliances). However, since dolphins could produce the reported results by only discriminating members of their own second-order alliance from individuals outside of that alliance, they may just track characteristics of their ingroup rather than a more nuanced social system or anything concept-like.

Author response: We respectfully disagree here and, to explain why, it is important to place our results in the context of the dolphin social system. Bottlenose dolphins do not defend territories or form stable social groups. Group composition can change on a minute-by-minute or hour-by-hour basis, and there is no sex-biased dispersal (both sexes are philopatric). This differs from most terrestrial mammals, which live in semi-closed, often territorial groups with one or more reproductive females, and where maturing males tend to disperse to other groups. These nested alliances are not comparable to neighbouring social groups or “in-group vs out-group”, as we see in most social birds and animals (we now try to make this clear in lines 54-63). In their nested network of alliances, where males across all three levels have known each other since birth (new Figure 2), it is not past nor contemporary bond strength (i.e., the time spent together) that drives their responses but whether or not they have supported each other historically in the pursuit and defence of females. They respond more strongly to males with whom they share a cooperative history. We argue that discriminating between rival groups is not nearly as striking a result as discriminating between “friends” based on cooperative history, which is far more nuanced.

A second point is that social concepts are generally thought to have predictive power. Conceiving of particular social relationships (mother-offspring, allies, etc) conceptually allows me to generate predictions about conspecifics' behavior as soon as I identify them as a member of a particular conceptual class. For example, researchers like Cheney and Seyfarth have presented a variety of manipulations that suggest that monkeys have expectations of social behavior that stem from such classification (e.g., monkeys respond more strongly to playbacks of rank reversals between matriline than rank reversals within matriline and more strongly to rank reversal within matriline than rank-consistent playbacks, indicating expectations about the likelihood and consequences of these different simulated social events). Because dolphins were discriminating individual signature whistles rather than more complex social information, it is not clear whether they have any expectation about how their second-order allies will behave relative to other individuals.

Author response: One could argue that discriminating between individuals (i.e., signature whistles) based on individual relationship histories is comparable in complexity to, perhaps even more complex than, grouping together individuals based on rank. Both can be explained by concept learning, but whereas rank reversals can be explained by relational concepts (more dominant vs less dominant), grouping individuals based on experience of individual histories can be explained by associative concepts. We now explain the three different types of concept learning in animals; perceptual, relational and associative, and explain how our results support associative concept learning (lines 300-312). We have also now added a sentence on future work that could investigate whether the social concept we

describe can be used predictively (lines 312-313).

More generally, conceptual understanding is a prominent topic in psychology, with major debates concerning whether young children and nonhuman animals are capable of any comprehension of concepts. Some (e.g., Seyfarth and Cheney, 2015, whom the authors reference) have proposed that certain species, like primates, are capable of generating basic and implicit conceptual knowledge while others have argued that the capacity to generate concepts is what separates humans from other species (Penn et al., 2008, Darwin's mistake: Explaining the discontinuity between human and nonhuman minds. Behavioral and Brain Sciences). If the authors wish to use the word 'concept' they should clearly define the term based on past literature and better defend its use.

Author response: We note here that Penn et al 2008 received a significant amount of criticism, including the 24 commentaries provided in response to the original paper. We have now clearly defined the three types of concept learning in animals; perceptual, relational and associative, and explain how our results fit the definition of associative concept learning (lines 300-312).

Analyses:

I also have some questions about the analyses. First, from the main text I had the impression that the authors were employing a traditional hypothesis testing approach in which a full model is generated with all factors necessary to test different hypotheses and account for necessary controls. However, the methods section makes clear that a model selection approach has been used instead. The authors should make this more explicit in the main text. Also, as far as I am aware, when employing a model selection approach, one should report the direction and strength of the effects in the best fitting model(s) using various terms but should not report p-values (as all factors included within the best-fitting models are considered important in explaining variation in the dependent measure).

Author response: Yes, we have followed the advice of all the reviewers in this and modified our statistical analyses accordingly. We now compare the full model with the null model (using a traditional hypothesis testing approach). In all cases, the full model explained significantly more variance than the null model (Table S1). While we now acknowledge that we employed a traditional hypothesis testing approach (lines 522-523), we also present backwards model selection tables in the ESM (Table S2) for those readers that are interested to know which of the fixed effects significantly added to the model.

Second, with regard to random effects, does 'playback target' refer to the subject? Subject ID should definitely be included as a random effect and it should be made clearer that this was the case. I would also be inclined to include as additional random effects the ID of the individual whose call has been played back and the second-order alliance of the subject. I appreciate that the limited sample size likely prohibited this possibility (since doing so would have made the models overly complex). However, since it would be appropriate to include such random effects, the authors should explicitly state in the methods that they could not include them as a result of sample size limitations.

Author response: Yes, playback target refers to the subject and we now refer to it as 'subject ID' throughout the manuscript. We also now include caller ID as a random effect; so, all the models have subject ID and caller ID included as random effects.

Related to the above points, it is hard to interpret statements such as the following (lines 203-204): "Interestingly, neither the current social bond strength nor the number of consortships in which the male played back and the target male(s) were partnered over time predicted response strength." Does this mean that those terms were not represented in the best fitting models or that they were in the best-fitting model but were not significant? This is consequential for interpreting the relationship between the various predictors relative to the analysis approach (i.e., how alternative explanations have been controlled for). In viewing the supplementary Model Selection Tables, I

find the reported effect confusing (that bond strength and consortships do not predict responses to the experimental manipulation) since the factor for consortships is present in the best-fitting model for response duration and the second best-fitting model for approach distance.

Author response: We have modified our analyses and now use a traditional hypothesis testing approach. Thus, the terms are represented in the full model but were not significant. We have now moved our results table that contains all the variables in the full model to the main text (Table 2).

The authors should also be clearer in the main text whether there was only one best-fitting model or multiple (which appears to be the case from the supplementary Model Selection Tables), and which factors were retained in the best-fitting model(s).

Author response: We have modified the manuscript to make it clear that we are using one best-fitting model (lines 521-522).

General Clarity:

A few details should be moved from the methods section into the main text to ensure that readers can clearly understand the general details of the study without having to consult the methods.

-First, the authors should clarify the manipulation itself. In line 124, I wasn't sure whether 'paired signature whistles of either their second- or third-order allies' meant (1) that dolphins participated in a within-subject design in which they heard a single signature whistle per trial from a member of their second-order alliance (condition 1) or third-order alliance (condition 2; i.e., paired conditions), or (2) whether dolphins heard two signature whistles of different individuals in close succession (i.e., paired whistles within a condition). As it turns out, checking the methods revealed that both of my inferences were wrong. Paired referred instead to back-to-back signature whistles from the same individual. However, even after reading the methods section, I am not clear on how many trials each subject experienced and whether the design was fully within-subject or a mixed within- and between-subject data set.

Author response: The study is a within-subject, repeated measures design in which males are exposed to both second- and third-order alliance playbacks. We now clarify this in lines 115-117. We have removed the term 'paired' to avoid confusion, and also provide our full data file with all this information as ESM.

-Second (e.g., lines 124-130), the authors should be explicit about what counts as a response. As far as I can tell, this was not described at all in the main text and so I could not clearly process the results that immediately followed.

Author response: We have now added this to lines 120-122.

In the introduction, the structure of male alliances is well explained but I think the reader would benefit from description of how various-order alliances relate to broader group structure for males and females. That is, for males, are second-order alliances the unit of a typical social group (e.g., with shared territory) but these groups sometimes engage cooperatively (in third-order alliances) or competitively with neighboring second-order alliance groups? At what level are females considered part of the social group, and how stable are their associations with one another and with various levels of alliances?

Author response: The dolphin social system differs markedly from those of most terrestrial mammals. Bottlenose dolphins do not form stable social groups (group composition can change on a minute-by-minute or hour-by-hour basis) and do not establish or defend territories. Thus, we cannot refer to 'neighbouring groups', since allied males and rivals share extensively overlapping ranges (ref 21, Randic et al. 2012) in an open social network with a highly dynamic fission-fusion grouping pattern. Individuals can form stable social relationships, with female association patterns influenced by range overlap and maternal

and biparental relatedness (ref 23, Frère et al 2010), and males forming long-term, multi-level alliances of typically unrelated individuals (e.g. refs 4, 7, 24, 71). We now make this much clearer in the introduction (lines 54-63).

Other Comments:

Line 85: Henceforth does not seem to be appropriate for this sentence

Author response: We have changed this word.

Overall, I find the work very interesting and if the authors could strongly support the claims that dolphins exhibit social concepts or that they can represent the nested structure of their alliance system, then I think the paper would definitely warrant publication in Nature Communications. However, as much as I would love to see this, at present, I'm not sure those claims can be satisfactorily defended with the available data. Because the findings only pertain to discrimination of second-order allies from others (who happen to be third-order allies) but no discrimination of any other levels (e.g., third-order allies from non-allies), the experiment cannot expose whether dolphins understand the nested structure of their alliance system. Whether the data could reflect a basic social concept depends most fundamentally, in my view, on whether the authors can exclude all possible dyadic traits that might co-vary with one's second-order alliance grouping and could explain more responsiveness to individuals who happen to be in one's second-order alliance. I appreciate the authors' effort to control for two such dyadic factors but I'm afraid their exclusion is not sufficient to confirm that another such variable isn't responsible. And without some additional evidence (e.g., discriminating between multiple social units that the subject is not part of, or generating expectations on the basis of classification), the evidence is insufficient to assume that dolphins categorize or group members of their second-order alliance in their minds.

Author response: We thank Reviewer 3 for the generally positive appraisal of our work. We have heeded the constructive criticism, refined our analyses and made significant additions to our interpretation and explanation of the striking results. We do now feel we have fully addressed these concerns in our above comments and the revised manuscript, and we summarise below:

- **We have provided a much clearer description of the dolphin social system, within which these results should be considered. Bottlenose dolphins do not defend territories or form stable social groups (group composition changes on a minute-by-minute or hourly basis), and there is no sex-biased dispersal as both sexes are philopatric. This differs from most terrestrial mammals, which live in semi-closed, often territorial groups with one or more reproductive female/s, and where maturing males tend to disperse to other groups. In their mosaic of overlapping home-ranges, Shark Bay's nested male alliances are not comparable to neighbouring social groups or "in-group vs out-group" as we see in many social birds and mammals (we have now made this clearer in lines 54-63).**
- **We provide the full 30-year dataset of dyadic relationship history, i.e., the time males have spent together, since they were calves and juveniles up until the present day/time of the study (this is visually portrayed in the new Figure 2).**
- **Third, we clarify that dolphins do not forage cooperatively in this population (lines 313-317) and that, if males are not foraging, they are spending their time herding females (lines 315-316). Thus, the time these males spend together (i.e. bond strength), the number of consortships they have participated in together and their cooperative support in the defence of females are the three traits that can define these alliances. Our results clearly show that bond strength and dyadic consortship history do not drive response strength. Instead, males distinguish second-order alliance members based on their historical cooperative support in the pursuit and defence of females.**
- **An important point that is missing here is that our results clearly show that males do not discriminate between first- and second-order allies, even though they spend far more time with some of their first-order allies and herd females with them.**

- **We argue that discriminating between rival groups is not nearly so striking a result as discriminating between “friends” (i.e., members of second and third order alliances) based on cooperative history (i.e., helping in the theft and defense of females), which is far more nuanced.**

REVIEWER COMMENTS

Reviewer #1 (Remarks to the Author):

I would like to congratulate the authors for an excellent job in the revision of the manuscript. They addressed most of the points and the text has improved a lot in clarity. The introduction and discussion are now more streamlined, and I especially enjoyed reading the new version of the discussion focus on associative learning instead of abstract concept formation. Regarding the methods, the authors clarified most of the points that we asked for and included new information about the dolphin social system that I believe will help the reader to understand the logic of the study. They changed their statistical approach from model selection to hypothesis testing, which is more suitable for their experimental design. However, there are still a few points that need to be addressed before publishing the paper.

INTRODUCTION

1. Thanks to some of the lines added in the text, I can now understand the dolphin social system. However, it is confusing that the authors first say that dolphins do not form stable social groups (authors response to comment 1 in the previous review and line 57 in the manuscript) but the main point of the manuscript is to explore whether the dolphins know that they belong to a stable social group -what else is a team? Whether they meet once a month or are together all the time does not matter, it is still a stable social group. What is the difference between a social group (line 57) and stable social relationships (line 59)

2. Another term that I find confusing is “cooperation-based concept”. I find the idea that dolphins understanding their team membership very interesting, and I agree with the authors that team membership is a concept based on cooperation. Cooperation is a very appealing term in research, but it has also a very wide range of meanings. It comes as a surprise that they speak about cooperation-based concepts and only in some lines they specify that they mean team membership. For example, the consortship history (that is also a form of cooperation) is not significant. I would rather go for calling it team membership-based concept (for example, in the title of the manuscript), and after discussing it (as they actually do) in the context of cooperation. Honestly, I don't think this change would make their findings less engaging at all.

METHODS

3. When we suggested that the presence of other individuals could potentially influence the response of the subjects (point 6), the authors replied that the response strength of the subject was either the same or even stronger than the other males present and that the subject responded first. They argued that this is enough to show that social facilitation cannot explain the result. However, I cannot see how social facilitation can be ruled out because of this. The subject could still be faster/stronger in their response because he knows others are looking. Interestingly, they also mention that in case a female was present, they seem to track the male's behaviour. If they know that a possible mate is paying attention to their actions, how can we be sure that the subjects are not changing their behaviour because they know they are being watched?

4. The statistical analysis is now much more appropriate, and I am glad that the authors take into account the reviewers' suggestion of hypothesis testing approach. There are still some aspects of their analysis that can be improved or clarified. The full-null model testing should be used to test a hypothesis. The authors follow the team membership hypothesis, that would predict that the PB type is what affects the variable response. Therefore, the null model should include all the other predictors being in the full model, lacking only the predictor of interest. What they did was to fit null models lacking all the predictors and containing only the intercept. In the view of the results, that are quite consistent, I am sure that their full-null comparison will still be significant independently of the predictors that they include in the model. That is, their results would not change at all, but still I fill that if they follow a hypothesis, their full-null model comparison should test for that hypothesis specifically, not for what all the predictors as a whole are adding to the model.

5. The authors nicely provide information about some assumptions of their model (e.g. they inspected the residuals and collinearity). They should also provide the model stability, that detects if there are influential cases in a given model. There are a number of ways to do it, for example using the package `influence.ME` in R (Nieuwenhuis, R.; Grotenhuis, H.F., and Pelzer, B.J., 2012).

6. As mentioned before, the use of AIC or any other type of model selection is not justified when you want to test a hypothesis (Lubke and Campbell, 2016; Mundry, 2011). Presenting it (even if it is only in the supplementary material) can create confusion and induce other researchers to think that it is an appropriate method in these cases. To avoid promoting bad scientific practices, it should be removed from the manuscript (main text and supplementary material).

7. I am not sure why in model 5 (orientation to source as response) they did not include all the other predictors (consort, SRI, PB order...) in the same way as in the rest of the models

DISCUSSION

8. In point 18, we asked why the authors think that familiarity is not affecting their results at all, even when there could potentially be an interaction effect between the strength of the social bonds and the belonging to the same second order alliance. They responded explaining that social bonds are a measure of how much time individuals are together and, since they included this in the model, it is supposed to be accounted for. They also mentioned that interactions effects occur when the effect of one variable depends on the value on another variable and, since alliance membership does not depend on bond strength, then they did not include the interaction in the model. However, interaction effects occur regarding the response variable. Even if social bonds are independent from alliance membership, it is possible that these two variables interact regarding the response variable (e.g. response duration). For example, social bound could affect the response only in the case of third order alliance (e.g. subjects look longer to the source if the call belongs to a third order alliance member that they are bonded with), but not be relevant in the case of second order alliance members. I am aware that testing for this effect is not the main aim of the authors, so they do not need to include the interaction in the model, but they need to be aware that, in the current form, the manuscript does not offer any data that allows to say

that familiarity does not play a role at all, unless they include the interaction in the model.

9. The authors have done a great job in the discussion, explaining why consortship might not have influenced their results in the light of team member cooperation. For me it is still puzzling why cooperation at the mate guarding level is not significant. Furthermore, experience with another dolphin is in the end a kind of familiarity, so it is difficult to see how familiarity is not playing a role. Anyhow, it is always difficult to understand exactly what the animals are thinking/perceiving about the world, and the authors have done a very good job in revising the manuscript and trying to discuss their results.

Minor comments

1. In the abstract the authors state that “they used 30 years of data... (together with playback experiments, etc.)”. There is nothing wrong with that, since they used these 30 years of data to describe some aspects of the social system of their subjects. However, the main conclusion of the manuscript is based on 40 playbacks on 14 males carried out in one year. For that reason, it would be better to provide the actual number of subjects of the study in order to prepare the reader for what they are going to find in the manuscript.

2. Regarding our point about the use of Welch’s test, I can now understand why they could not use paired t-test. However, most of the data is still paired data, so the assumption of unequal variances is not so clear to me... An alternative could be to use a survival analysis like the Cox regression model (Jahn-Eimermacher, A., Lasarzik, I., and Raber, J., 2011) in the survival package in R (Therneau, T.M., 2015).

Reviewer #2 (Remarks to the Author):

The authors have done a very good job and submitted a much improved revised version of the manuscript. Congratulations to a fine contribution to the literature! I only have three minor issues that require attention. First, Figure 3 should not combine a barplot with the raw data, but a line indicating the mean with the measure of variation (please indicate whether this is the CI or the SD or whatever, also in the Figure legend). Barplots should *only* be used for proportions or counts. Second, I strongly discourage the presentation of Tables S1 and S2. They give a feel of data dredging and do not provide any additional critical information. Third, line 78 ff - this sentence makes for a hard read. Perhaps insert "that is" before "significant association preferences".

Reviewer #3 (Remarks to the Author):

I commend the authors on their thoughtful revision and response to reviews. I find the paper much clearer, the statistics much stronger, and the organization improved. I also appreciate that the authors build up to use of the term concept, favoring more direct description of the evidence earlier on as well as use of the term classify (which although somewhat loaded is much less loaded than concept), and that they offer more detailed discussion of cognitive and affective/endocrinological mechanisms in the

discussion. This is an exciting and highly novel study, integrating impressive and extremely hard-to-generate datasets, to test important questions about the social lives and social cognition of dolphins. It is sure to have a pronounced impact. Open questions remain about exactly how dolphins cognitively represent the social world but these open questions should not temper the importance of this work (and instead point to exciting future directions). Provided that the authors would be willing to make the below additional changes, I would be very happy to recommend the paper for publication.

Discussion Points:

1. I agree with reviewer 1 that the authors should attempt to further address alternative explanations.

1a. The hormonal mechanism described in lines 289-298 provides an interesting and important potential explanation for dolphins' discrimination of second-order allies from others. However, the authors should explicitly note that while endocrine activity could reinforce associative concepts and/or facilitate concept learning, hormonal mechanisms could also represent an alternative explanation to the cognitive one that the authors favor. Since the test metric is approach and/or duration of interest (rather than an explicit grouping, as in a touchscreen task, or some metric of prediction or violation of expectation), the differential response in these metrics could, in principle, be underlain solely by the sorts of affective bonding mechanisms that the authors describe (without the need for any cognitive concepts). The authors do not need to favor this low-level explanation but they should note it as potentially compatible with OR an alternative to the cognitive one.

1b. Regarding familiarity or first-order alliances, one possibility is that association or consortship rates do not predict responses to playbacks because the data do not accurately reflect relative association or consortship rates. The authors should raise the possibility that various sampling biases (e.g., animals are not constantly tracked so data offer only partial coverage; some animals may be better habituated or more central in the social network and thus more often observed, etc) could impact their results, and explain why such biases are unlikely to do so (e.g., these same data have demonstrated predictive power in other studies, etc). For example, if association data are used to classify second-order allies, is it possible that those data are sufficiently accurate to identify the strongest relationships categorically but not accurate enough to characterize more subtle variation (e.g., amongst strongly associated animals)? This would be one reason why the second vs third-order alliance predictor would be significant rather than the predictor continuously quantifying dyadic association (and given that these predictors are based on similar data, they presumably covary such that both are unlikely to result in independent, significant effects).

2. The authors should also note that, because their results suggest a stronger response to second-order allies than to one particular other group (i.e., third-order allies), it remains an open question how complex dolphins' representation of their social structure is. (2a) They may, for example, respond selectively to second-order allies relative to all other groups (e.g., stronger response to second-order allies than to third-order allies or to non-allies, and potentially no difference in responses to third-order allies vs non-allies). (2b) It also remains an open question whether dolphins represent their dyadic relations with second-order allies in a privileged way (i.e., each individual relation being distinguished

from the subject's other relations) or whether they additionally group second-order allies into a conceptual category. These are interesting future directions that should be acknowledged.

Introduction: In the literature on social knowledge, there is a major distinction between tracking one's own direct first-person relationships (the focus of the present work) and tracking third-party relationships. In fact, most work on social concepts has focused on third-party relationships (e.g., most work by Cheney and Seyfarth) and generally I think many readers might assume that social concept refers to third-party relationships. For this reason, I think it is important that the authors are very clear (more than they currently are) that the focus of their work is on classification of dolphins' own first-person relationships.

Title: The authors' evidence is consistent with concept formation but, as described above, alternative explanations remain (e.g., an affective/hormonal mechanism that does not require cognitive representation of associative concepts). As such, I think the title should be adjusted: either made more descriptive (e.g., "Male bottlenose dolphins discriminate conspecifics according to first-person alliance levels") or at least toned down somewhat (e.g., "Evidence of first-person cooperation-based concept formation in male bottlenose dolphins").

Reviewer #1 (Remarks to the Author):

I would like to congratulate the authors for an excellent job in the revision of the manuscript. They addressed most of the points and the text has improved a lot in clarity. The introduction and discussion are now more streamlined, and I especially enjoyed reading the new version of the discussion focus on associative learning instead of abstract concept formation. Regarding the methods, the authors clarified most of the points that we asked for and included new information about the dolphin social system that I believe will help the reader to understand the logic of the study. They changed their statistical approach from model selection to hypothesis testing, which is more suitable for their experimental design. However, there are still a few points that need to be addressed before publishing the paper.

Author response: We would like to thank the reviewer for their thoughtful and constructive comments on our manuscript. We provide details on how we have addressed the remaining points below.

INTRODUCTION

1. Thanks to some of the lines added in the text, I can now understand the dolphin social system. However, it is confusing that the authors first say that dolphins do not form stable social groups (authors response to comment 1 in the previous review and line 57 in the manuscript) but the main point of the manuscript is to explore whether the dolphins know that they belong to a stable social group -what else is a team? Whether they meet once a month or are together all the time does not matter, it is still a stable social group. What is the difference between a social group (line 57) and stable social relationships (line 59)

Author response: The point we are conveying here is that these social “groups”, i.e. alliances, are not closed, as they are in many social birds and mammals. These males have long-term social relationships with each other while still residing in an open social network. We have added an additional line to the introduction (line 57-65) to make this clearer:

“In this system, young males are not part of a closed social unit from which to choose alliance partners, instead they develop long-term bonds with other males within this open social network.”

2. Another term that I find confusing is “cooperation-based concept”. I find the idea that dolphins understanding their team membership very interesting, and I agree with the authors that team membership is a concept based on cooperation. Cooperation is a very appealing term in research, but it has also a very wide range of meanings. It comes as a surprise that they speak about cooperation-based concepts and only in some lines they specify that they mean team membership. For example, the consortship history (that is also a form of cooperation) is not significant. I would rather go for calling it team membership-based concept (for example, in the title of the manuscript), and after discussing it (as they actually do) in the context of cooperation. Honestly, I don't think this change would make their findings less engaging at all.

Author response: Our results show that male dolphins classify / identify team members using an associative concept based on cooperative history. This concept is based on cooperation (as the reviewer states above) and we feel strongly that our current title clearly summarises our results and is readily intelligible to any scientist. We would therefore much prefer to keep our current title.

While direct consortship history was not significant, we discuss in the manuscript that, while the number of consortships did not explain response strength, consortships do occur exclusively within the second-order alliance. Even if male A has not consorted a female with male B, the triadic interactions within second-order alliances mean that it is likely that A has consorted a female with another male (e.g. C) that, in turn, has consorted a female with B.

Investing in, and responding to, all second-order alliance members equally may therefore result in significant by-product benefits.

METHODS

3. When we suggested that the presence of other individuals could potentially influence the response of the subjects (point 6), the authors replied that the response strength of the subject was either the same or even stronger than the other males present and that the subject responded first. They argued that this is enough to show that social facilitation cannot explain the result. However, I cannot see how social facilitation can be ruled out because of this. The subject could still be faster/stronger in their response because he knows others are looking. Interestingly, they also mention that in case a female was present, they seem to track the male's behaviour. If they know that a possible mate is paying attention to their actions, how can we be sure that the subjects are not changing their behaviour because they know they are being watched?

Author response: We now present an analysis in the Supplementary Information to justify not including group size in our model (and thus not over complicating our analyses), and we refer to this new analysis in the main text (lines 520-522). We used t-tests to show there is no difference in either response duration or approach distance for playbacks to groups versus those to individuals. We ran this analysis separately for second- and third-order playbacks to ensure the presence of group members was not influencing the response of individuals to one playback treatment and not the other.

4. The statistical analysis is now much more appropriate, and I am glad that the authors take into account the reviewers' suggestion of hypothesis testing approach. There are still some aspects of their analysis that can be improved or clarified. The full-null model testing should be used to test a hypothesis. The authors follow the team membership hypothesis, that would predict that the PB type is what affects the variable response. Therefore, the null model should include all the other predictors being in the full model, lacking only the predictor of interest. What they did was to fit null models lacking all the predictors and containing only the intercept. In the view of the results, that are quite consistent, I am sure that their full-null comparison will still be significant independently of the predictors that they include in the model. That is, their results would not change at all, but still I fill that if they follow a hypothesis, their full-null model comparison should test for that hypothesis specifically, not for what all the predictors as a whole are adding to the model.

Author response: We have now made this comparison where our null model contains all the fixed and random effects, except the fixed effect of interest (PB type). The full-null model comparisons remain significant and we have updated Table S1.

5. The authors nicely provide information about some assumptions of their model (e.g. they inspected the residuals and collinearity). They should also provide the model stability, that detects if there are influential cases in a given model. There are a number of ways to do it, for example using the package `influence.ME` in R (Nieuwenhuis, R.; Grotenhuis, H.F., and Pelzer, B.J., 2012).

Author response: Thanks for this recommendation. We have now calculated Cook's distances of single observations for both models using the `influence.ME` package in R and detected no influential cases (all values < 1) ¹. See lines 536-537.

6. As mentioned before, the use of AIC or any other type of model selection is not justified when you want to test a hypothesis (Lubke and Campbell, 2016; Mundry, 2011). Presenting it (even if it is only in the supplementary material) can create confusion and induce other researchers to think that it is an appropriate method in these cases. To avoid promoting bad scientific practices, it should be removed from the manuscript (main text and supplementary material).

Author response: We have now removed Table S2 (backwards model selection) from the main text and the manuscript.

7. I am not sure why in model 5 (orientation to source as response) they did not include all the other predictors (consort, SRI, PB order...) in the same way as in the rest of the models

Author response: We now include the same fixed and random effects as the rest of the models.

DISCUSSION

8. In point 18, we asked why the authors think that familiarity is not affecting their results at all, even when there could potentially be an interaction effect between the strength of the social bonds and the belonging to the same second order alliance. They responded explaining that social bonds are a measure of how much time individuals are together and, since they included this in the model, it is supposed to be accounted for. They also mentioned that interactions effects occur when the effect of one variable depends on the value on another variable and, since alliance membership does not depend on bond strength, then they did not include the interaction in the model. However, interaction effects occur regarding the response variable. Even if social bonds are independent from alliance membership, it is possible that these two variables interact regarding the response variable (e.g. response duration). For example, social bond could affect the response only in the case of third order alliance (e.g. subjects look longer to the source if the call belongs to a third order alliance member that they are bonded with), but not be relevant in the case of second order alliance members. I am aware that testing for this effect is not the main aim of the authors, so they do not need to include the interaction in the model, but they need to be aware that, in the current form, the manuscript does not offer any data that allows to say that familiarity does not play a role at all, unless they include the interaction in the model.

Author response: We would define familiarity as 'how well you know someone', i.e. how familiar you are with them. This can be measured by bond strength in our study (and many others), which is a measure of the proportion of time two individuals spend together. This variable was not significant in our models. Furthermore, we included Figure 2 to show that all males in our study have spent time together since they were calves and juveniles, well before they formed their second-order alliances. Thus, they have all been familiar with each other for decades. We therefore feel that we have adequately addressed the issue of familiarity.

Nevertheless, we decided to compare our full models (as presented in the manuscript) with a model with the same fixed and random effects but with an interaction term between bond strength and alliance membership. For all response measures (response duration, approach distance and orient to source), the model with the interaction term was not favoured over our current full model (nor was the term significant). We subsequently dropped the term from our models to avoid over-complicating our analyses. We now state this in the methods (lines 533-535).

9. The authors have done a great job in the discussion, explaining why consortship might not have influenced their results in the light of team member cooperation. For me it is still puzzling why cooperation at the mate guarding level is not significant. Furthermore, experience with another dolphin is in the end a kind of familiarity, so it is difficult to see how familiarity is not playing a role. Anyhow, it is always difficult to understand exactly what the animals are thinking/perceiving about the world, and the authors have done a very good job in revising the manuscript and trying to discuss their results.

Author response: Thank you. The results are indeed interesting. As we mention earlier, we discuss in the manuscript that, while the number of consortships did not explain response strength, consortships do occur almost exclusively within the second-order alliance. Even if male A has not consorted a female with male B, the triadic interactions within second-order alliances mean that it is likely that A has consorted a female with another male (e.g. C) that, in turn, has consorted a female with B. Investing in, and responding to, all second-order alliance members equally may therefore result in significant by-product benefits (lines 268-273). We

also acknowledge that the responses we recorded should be considered low cost, so it remains to be determined if higher cost investments accord with bond strength or consortship history within second-order alliances (lines 338-340). However, successful fights against rival alliances allow males to keep their female or acquire one. Thus, supporting, and receiving support from, second-order allies significantly contribute to a male's lifetime reproductive success (lines 332-334).

Minor comments

1. In the abstract the authors state that "they used 30 years of data... (together with playback experiments, etc.)". There is nothing wrong with that, since they used these 30 years of data to describe some aspects of the social system of their subjects. However, the main conclusion of the manuscript is based on 40 playbacks on 14 males carried out in one year. For that reason, it would be better to provide the actual number of subjects of the study in order to prepare the reader for what they are going to find in the manuscript.

Author response: We have now added the number of subjects (14) and the number of playbacks (40) in the abstract.

2. Regarding our point about the use of Welch's test, I can now understand why they could not use paired t-test. However, most of the data is still paired data, so the assumption of unequal variances is not so clear to me... An alternative could be to use a survival analysis like the Cox regression model (Jahn-Eimermacher, A., Lasarzik, I., and Raber, J., 2011) in the survival package in R (Therneau, T.M., 2015).

Author response: This was a small part of our analysis that tells us if the mean time to response between the two playback types was significantly different. The assumption of unequal variances is automated in R. We would like to leave the test as is, with the acknowledgement that we can't generalize this finding beyond our experiments (as we aren't accounting for repeated measures). Alternatively, we could remove the test and just use descriptive statistics (i.e. mean +/- sd of latency to response for the two playback types).

Reviewer #2 (Remarks to the Author):

The authors have done a very good job and submitted a much improved revised version of the manuscript. Congratulations to a fine contribution to the literature! I only have three minor issues that require attention. First, Figure 3 should not combine a barplot with the raw data, but a line indicating the mean with the measure of variation (please indicate whether this is the CI or the SD or whatever, also in the Figure legend). Barplots should *only* be used for proportions or counts. Second, I strongly discourage the presentation of Tables S1 and S2. They give a feel of data dredging and do not provide any additional critical information. Third, line 78 ff - this sentence makes for a hard read. Perhaps insert "that is" before "significant association preferences".

Author response: We would like to thank the reviewer for their thoughtful and constructive comments on our manuscript. We have now modified figure 3 as suggested (where black dots represent the mean and the line represents the 95% confidence intervals). We have removed Table S2, but have kept Table S1 because it contains the results of the full vs null model comparison. We have modified line 78 to improve clarity.

Reviewer #3 (Remarks to the Author):

I commend the authors on their thoughtful revision and response to reviews. I find the paper much clearer, the statistics much stronger, and the organization improved. I also appreciate that the authors build up to use of the term concept, favoring more direct description of the evidence earlier on as well as use of the term classify (which although somewhat loaded is much less loaded than concept), and that they offer more detailed discussion of cognitive and affective/endocrinological mechanisms in the discussion. This is an exciting and highly novel study, integrating impressive and extremely hard-to-

generate datasets, to test important questions about the social lives and social cognition of dolphins. It is sure to have a pronounced impact. Open questions remain about exactly how dolphins cognitively represent the social world but these open questions should not temper the importance of this work (and instead point to exciting future directions). Provided that the authors would be willing to make the below additional changes, I would be very happy to recommend the paper for publication.

Author response: We would like to thank the reviewer for their thoughtful and constructive comments on our manuscript. We provide details on how we have addressed the remaining points below.

Discussion Points:

1. I agree with reviewer 1 that the authors should attempt to further address alternative explanations.

Author response: We now provide more information on alternative explanations below (1a and 1b).

1a. The hormonal mechanism described in lines 289-298 provides an interesting and important potential explanation for dolphins' discrimination of second-order allies from others. However, the authors should explicitly note that while endocrine activity could reinforce associative concepts and/or facilitate concept learning, hormonal mechanisms could also represent an alternative explanation to the cognitive one that the authors favor. Since the test metric is approach and/or duration of interest (rather than an explicit grouping, as in a touchscreen task, or some metric of prediction or violation of expectation), the differential response in these metrics could, in principle, be underlain solely by the sorts of affective bonding mechanisms that the authors describe (without the need for any cognitive concepts). The authors do not need to favor this low-level explanation but they should note it as potentially compatible with OR an alternative to the cognitive one.

Author response: We now make this point clearer in lines 299-301 and lines 352-353.

1b. Regarding familiarity or first-order alliances, one possibility is that association or consortship rates do not predict responses to playbacks because the data do not accurately reflect relative association or consortship rates. The authors should raise the possibility that various sampling biases (e.g., animals are not constantly tracked so data offer only partial coverage; some animals may be better habituated or more central in the social network and thus more often observed, etc) could impact their results, and explain why such biases are unlikely to do so (e.g., these same data have demonstrated predictive power in other studies, etc). For example, if association data are used to classify second-order allies, is it possible that those data are sufficiently accurate to identify the strongest relationships categorically but not accurate enough to characterize more subtle variation (e.g., amongst strongly associated animals)? This would be one reason why the second vs third-order alliance predictor would be significant rather than the predictor continuously quantifying dyadic association (and given that these predictors are based on similar data, they presumably covary such that both are unlikely to result in independent, significant effects).

Author response: We respectfully disagree that this warrants further consideration in the manuscript. We have previously clarified that our sightings data lead to robust network metrics, including bond strength, based on simulations we have conducted with animals where hundreds of sightings are available – we have shown that a minimum number of 15 sightings is sufficient to build robust social networks, as per numerous previous studies. We detail in the manuscript that recent consortships and full consortship history are highly correlated, thus the pattern of consortships among males is consistent over years, which is evidence against incomplete sampling. Further, these three alliances have been extremely well-studied over the last 30 years, with hundreds of hours of focal follow data collected on all alliance members. The detail and long-term coverage we provide on these males represents one of the very strengths of this study.

Bottlenose dolphins exhibit a fission-fusion grouping pattern, in which group membership changes on a minute-by-minute or hourly basis, the study population is habituated to small research boats and they do not have a marked diurnal pattern of behaviour. There is therefore no reason to assume that their associations and behaviour are going to be markedly different between the hours, months and years over which we have observed them and chronicled their behaviour and those that we have not. We strongly feel that our data accurately reflect relative association and consortship rates, as documented in the last three decades of scientific publications, referenced throughout the manuscript.

2. The authors should also note that, because their results suggest a stronger response to second-order allies than to one particular other group (i.e., third-order allies), it remains an open question how complex dolphins' representation of their social structure is. (2a) They may, for example, respond selectively to second-order allies relative to all other groups (e.g., stronger response to second-order allies than to third-order allies or to non-allies, and potentially no difference in responses to third-order allies vs non-allies). (2b) It also remains an open question whether dolphins represent their dyadic relations with second-order allies in a privileged way (i.e., each individual relation being distinguished from the subject's other relations) or whether they additionally group second-order allies into a conceptual category. These are interesting future directions that should be acknowledged.

Author response: We now mention (2a) in lines 343-345. We feel we have already addressed (2b), as our main results show that males group second-order allies into a conceptual category, plus we already state that the responses we recorded should be considered low cost, so it remains to be determined if higher cost investments accord with bond strength within second-order alliances (lines 338-340).

Introduction: In the literature on social knowledge, there is a major distinction between tracking one's own direct first-person relationships (the focus of the present work) and tracking third-party relationships. In fact, most work on social concepts has focused on third-party relationships (e.g., most work by Cheney and Seyfarth) and generally I think many readers might assume that social concept refers to third-party relationships. For this reason, I think it is important that the authors are very clear (more than they currently are) that the focus of their work is on classification of dolphins' own first-person relationships.

Author response: We now make this distinction much clearer, both in the abstract (line 36), as well in the discussion (lines 313-317).

Title: The authors' evidence is consistent with concept formation but, as described above, alternative explanations remain (e.g., an affective/hormonal mechanism that does not require cognitive representation of associative concepts). As such, I think the title should be adjusted: either made more descriptive (e.g., "Male bottlenose dolphins discriminate conspecifics according to first-person alliance levels") or at least toned down somewhat (e.g., "Evidence of first-person cooperation-based concept formation in male bottlenose dolphins").

Author response: As the reviewer acknowledges, our evidence is consistent with concept formation and we strongly feel that our current title simply and effectively conveys our main findings. We would therefore much prefer to keep our current title. We have, however, modified our abstract to account for the reviewer's suggestions, including the distinction between first-person and third-person cooperation.

REVIEWERS' COMMENTS

Reviewer #2 (Remarks to the Author):

The authors have done an excellent job in addressing the reviewers' comments. Congratulations on a very interesting contribution to the field of social knowledge.

Reviewer #3 (Remarks to the Author):

I am satisfied with the authors' edits.

I still believe that it is possible that dolphins have privileged cognitive representations of second-order allies without conceptually grouping those allies together (reviewer 3 point 2b from the last round of reviews). This is in much the same way that animals may consistently respond differently to alpha-ranked individuals (and may have a representation of each individual alpha as supremely powerful, or whatever) without mentally representing all alphas as being part of a special conceptual class. It would be my preference that the authors do note this point in their discussion but regardless I am happy to see this exciting study published.

Reviewer #4 (Remarks to the Author):

I want to congratulate the authors for an excellent job in the revision of the manuscript. All the points were nicely addressed. The statistics are now suitable for the study hypothesis and design. Regarding the use of the Welch test, I agree with the author's response. This test does not involve the main hypothesis of the study and just test a minor question. Therefore, I am happy if they keep it if they acknowledge that they are not accounting for repeated measures. I think the manuscript still raises some question about what can be called a cooperation-based concept and what cannot, but the authors nicely discussed their point of view. I consider this a strength of the manuscript because it can generate fruitful scientific discussion. I think that the manuscript in its current form will be a nice addition to the literature about cooperation.

Reviewer #2 (Remarks to the Author):

The authors have done an excellent job in addressing the reviewers' comments. Congratulations on a very interesting contribution to the field of social knowledge.

Reviewer #3 (Remarks to the Author):

I am satisfied with the authors' edits.

I still believe that it is possible that dolphins have privileged cognitive representations of second-order allies without conceptually grouping those allies together (reviewer 3 point 2b from the last round of reviews). This is in much the same way that animals may consistently respond differently to alpha-ranked individuals (and may have a representation of each individual alpha as supremely powerful, or whatever) without mentally representing all alphas as being part of a special conceptual class. It would be my preference that the authors do note this point in their discussion but regardless I am happy to see this exciting study published.

Author response: We have now added a sentence on this in the discussion.

Reviewer #4 (Remarks to the Author):

I want to congratulate the authors for an excellent job in the revision of the manuscript. All the points were nicely addressed. The statistics are now suitable for the study hypothesis and design. Regarding the use of the Welch test, I agree with the author's response. This test does not involve the main hypothesis of the study and just test a minor question. Therefore, I am happy if they keep it if they acknowledge that they are not accounting for repeated measures. I think the manuscript still raises some question about what can be called a cooperation-based concept and what cannot, but the authors nicely discussed their point of view. I consider this a strength of the manuscript because it can generate fruitful scientific discussion. I think that the manuscript in its current form will be a nice addition to the literature about cooperation.

Author response: We now acknowledge that we did not account for repeated measures in that specific test.